# Oryx: a Scalable Sequence Model for Many-Agent Coordination in Offline MARL

**Claude Formanek**[*,1,2]    **Omayma Mahjoub**[1]    **Louay Ben Nessir**[1]    **Sasha Abramowitz**[1]
**Ruan de Kock**[1]    **Wiem Khlifi**[1]    **Daniel Rajaonarivonivelomanantsoa**[1,3]    **Simon Du Toit**[1]
**Arnol Fokam**[1]    **Siddarth Singh**[1]    **Ulrich Mbou Sob**[1]    **Felix Chalumeau**[1]
**Arnu Pretorius**[1,3]

[1]InstaDeep    [2]University of Cape Town    [3]Stellenbosch University

## Abstract

A key challenge in offline multi-agent reinforcement learning (MARL) is achieving effective many-agent multi-step coordination in complex environments. In this work, we propose **Oryx**, a novel algorithm for offline cooperative MARL to directly address this challenge. Oryx adapts the recently proposed retention-based architecture Sable (Mahjoub et al., 2025) and combines it with a sequential form of implicit constraint Q-learning (ICQ) (Yang et al., 2021), to develop a novel offline autoregressive policy update scheme. This allows Oryx to solve complex coordination challenges while maintaining temporal coherence over long trajectories. We evaluate Oryx across a diverse set of benchmarks from prior works—SMAC, RWARE, and Multi-Agent MuJoCo—covering tasks of both discrete and continuous control, varying in scale and difficulty. **Oryx achieves state-of-the-art performance on more than 80% of the 65 tested datasets**, outperforming prior offline MARL methods and demonstrating robust generalisation across domains with many agents and long horizons. Finally, we introduce new datasets to push the limits of many-agent coordination in offline MARL, and demonstrate Oryx's superior ability to scale effectively in such settings.

## 1  Introduction

Cooperative Multi-Agent Reinforcement Learning (MARL) holds significant potential across diverse real-world domains, including autonomous driving (Cornelisse et al., 2025), warehouse logistics (Krnjaic et al., 2024), satellite assignment (Holder et al., 2025), and intelligent rail network management (Schneider et al., 2024). Yet, deploying MARL in realistic settings remains challenging, as learning effective multi-agent policies typically requires extensive and costly environment interaction. This limits applicability to safety-critical or economically constrained domains, where the cost of real-world experimentation is prohibitively high. Fortunately, in many such settings, large volumes of logged data—such as historical train schedules, traffic records, or robot navigation trajectories—are available. By developing effective methods to distil robust, coordinated policies from these static datasets, we may unlock their full potential.

Offline MARL aims to address this exact challenge, training multi-agent policies solely from pre-collected data without further environment interaction. However, learning in the offline multi-agent setting introduces two primary difficulties. The first, well-studied challenge is *accumulating extrapolation error*, which occurs when agents select actions during training that fall outside the distribution of the offline dataset. This issue compounds rapidly as the joint action space grows

---

[*]Correspondence: c.formanek@instadeep.com

39th Conference on Neural Information Processing Systems (NeurIPS 2025).

exponentially with the number of agents (Yang et al., 2021). Recent works (Wang et al., 2023; Matsunaga et al., 2023; Shao et al., 2023; Bui et al., 2025) have made progress on addressing extrapolation error through policy constraints or conservative value estimation. However, such methods were typically only tested on settings with relatively few agents. Leaving open the question of whether such methods are able to scale to more agents.

The second issue is *miscoordination*, arising from the inability of agents to actively interact in the environment. Offline training forces agents to rely entirely on historical behaviours observed in the dataset collected from other (often suboptimal) policies, risking the development of incompatible policies. Tilbury et al. (2024) highlight how this miscoordination problem can significantly degrade performance in cooperative settings. Some recent works have tried to address this problem (Barde et al., 2024; Qiao et al., 2025). However, it's unclear how well these approaches scale, particularly when long temporal dependencies and many agents are involved.

To jointly tackle the two fundamental challenges of extrapolation error and miscoordination in many-agent settings, we propose **Oryx**, a novel offline MARL algorithm that unifies scalable sequence modelling with constrained offline policy improvement. Oryx integrates the retention-based sequence modelling architecture of Sable (Mahjoub et al., 2025) with an enhanced offline multi-agent objective based on ICQ (Yang et al., 2021). Furthermore, by leveraging a dual-decoder architecture—simultaneously predicting actions and Q–values—Oryx is able to use a counterfactual advantage (Foerster et al., 2018), enabling robust and extrapolation-safe policy updates. Finally, Oryx's sequential policy updating scheme explicitly addresses miscoordination by conditioning each agent's policy update on the actions already executed by other agents in sequence, thus ensuring stable policy improvement.

We extensively evaluate Oryx across a broad set of challenging benchmarks—SMAC, RWARE and Multi-Agent MuJoCo—that include discrete and continuous control, varying episode lengths, and diverse agent densities. Oryx sets a new state-of-the-art in offline MARL, outperforming existing approaches in more than 80% of the 65 evaluated datasets. Furthermore, we specifically test Oryx in large many-agent settings. To achieve this, we create new datasets (with up to 50 agents) and show that Oryx maintains its superior performance at such scales. Overall, by addressing both extrapolation error and miscoordination in a scalable sequence modelling framework, Oryx significantly advances offline MARL, bringing us closer to being able to reliably deploy cooperative, multi-agent policies learned entirely from static data in complex, real-world domains. To accelarate future research in this direction, we make all of our datasets and code available on GitHub[1].

## 2  Background

**Preliminaries – problem formulation and notation.** We model cooperative MARL as a Dec-POMDP (Kaelbling et al., 1998) specified by the tuple $\langle \mathcal{N}, \mathcal{S}, \mathcal{A}, P, R, \{\Omega^i\}_{i \in \mathcal{N}}, \{E_i\}_{i \in \mathcal{N}}, \gamma \rangle$. At each timestep $t$, the system is in state $s_t \in \mathcal{S}$. Each agent $i \in \mathcal{N}$ selects an action $a_t^i \in \mathcal{A}^i$, based on its local action-observation history $\tau_t^i = (o_0^i, a_0^i, \ldots, o_t^i)$, contributing to a joint action $\boldsymbol{a}_t \in \mathcal{A} = \prod_{i \in N} \mathcal{A}^i$. Executing $\boldsymbol{a}_t$ in the environment gives a shared reward $r_t = R(s_t, \boldsymbol{a}_t)$, transitions the system to a new state $s_{t+1} \sim P(\cdot|s_t, \boldsymbol{a}_t)$, and provides each agent $i$ with a new observation $o_{t+1}^i \sim E_i(\cdot|s_{t+1}, \boldsymbol{a}_t)$ to update its history as $\tau_{t+1}^i = (\tau_t^i, a_t^i, o_{t+1}^i)$. The goal is to learn a joint policy $\pi(\boldsymbol{a}|\boldsymbol{\tau})$, that maximizes the expected sum of discounted rewards, $J(\boldsymbol{\pi}) = \mathbb{E}_\pi \left[ \sum_{t=0}^\infty \gamma^t r_t \right]$.

In our work, we adopt the multi-agent notation from Zhong et al. (2024b). Specifically, let $i_{1:m}$ denote an ordered subset $\{i_1, \ldots, i_m\}$ of $\mathcal{N}$, then $-i_{1:m}$ refers to its complement, and for $m = 1$, we have $i$ and $-i$, respectively. The $k^{th}$ agent in the ordered subset is indexed as $i_k$. The action-value function is then defined as

$$Q^{i_{1:m}}(\boldsymbol{\tau}, \mathbf{a}^{i_{1:m}}) = \mathbb{E}_{\mathbf{a}^{-i_{1:m}} \sim \pi^{-i_{1:m}}} [Q(\boldsymbol{\tau}, \mathbf{a}^{i_{1:m}}, \mathbf{a}^{-i_{1:m}})].$$

Here, $Q^{i_{1:m}}(\boldsymbol{\tau}, \mathbf{a}^{i_{1:m}})$ evaluates the value of agents $i_{1:m}$ taking actions $\mathbf{a}^{i_{1:m}}$ having observed $\boldsymbol{\tau}$ while marginalizing out $\mathbf{a}^{-i_{1:m}}$. When $m = n$ (the joint action), then $i_{1:n} \in \mathrm{Sym}(n)$, where $\mathrm{Sym}(n)$ denotes the set of permutations of integers $1, \ldots, n$, which results in $Q^{i_{1:n}}(\boldsymbol{\tau}, \mathbf{a}^{i_{1:n}})$ being equivalent to $Q(\boldsymbol{\tau}, \mathbf{a})$. When $m = 0$, the function takes the form of canonical state-value function $V(\boldsymbol{\tau})$. Moreover, consider two disjoint subsets of agents, $j_{1:k}$ and $i_{1:m}$. Then, the multi-agent advantage

---

[1] https://github.com/instadeepai/og-marl

function of $i_{1:m}$ with respect to $j_{1:k}$ is defined as

$$A^{i_{1:m}}(\boldsymbol{\tau}, \mathbf{a}^{j_{1:k}}, \mathbf{a}^{i_{1:m}}) = Q^{j_{1:k}, i_{1:m}}(\boldsymbol{\tau}, \mathbf{a}^{j_{1:k}}, \mathbf{a}^{i_{1:m}}) - Q^{j_{1:k}}(\boldsymbol{\tau}, \mathbf{a}^{j_{1:k}}). \quad (1)$$

The advantage function $A^{i_{1:m}}(\boldsymbol{\tau}, \mathbf{a}^{j_{1:k}}, \mathbf{a}^{i_{1:m}})$ evaluates the advantage of agents $i_{1:m}$ taking actions $\mathbf{a}^{i_{1:m}}$ having observed $\boldsymbol{\tau}$ and given the actions taken by agents $j_{1:k}$ are $\mathbf{a}^{j_{1:k}}$, with the rest of the agents' actions marginalized out in expectation.

**Heterogeneous Agent RL framework – principled algorithm design for MARL.** Zhong et al. (2024b) show how practical and performant multi-agent policy iteration algorithms can be designed by leveraging sequential policy updates. Underlying much of this work is the multi-agent advantage decomposition theorem (Kuba et al., 2021), which states that

$$A(\boldsymbol{\tau}, \mathbf{a}) = A^{i_{1:n}}(\boldsymbol{\tau}, \mathbf{a}^{i_{1:n}}) = \sum_{j=1}^{n} A^{i_j}(\boldsymbol{\tau}, \mathbf{a}^{i_{1:j-1}}, a^{i_j}). \quad (2)$$

In essence, the above theorem ensures that if a policy iteration algorithm is able to update its policy sequentially across agents while maintaining positive advantage, i.e. $A^{i_j}(\boldsymbol{\tau}, \mathbf{a}^{i_{1:j-1}}, a^{i_j}) > 0 \,\forall j$, it guarantees monotonic improvement. This result has guided the design of several recent algorithms under the *heterogeneous agent RL* framework (Zhong et al., 2024b), including multi-agent variants of PPO (Kuba et al., 2022a), MADDPG (Kuba et al., 2022b) and SAC (Liu et al., 2024a).

**Sable – efficient sequence modelling with long-context memory.** Sable (Mahjoub et al., 2025) is a recently proposed online, on-policy sequence model for MARL. It is specifically designed for environments with long-term dependencies and large agent populations. Its key component is the *retention mechanism*, inspired by RetNet (Sun et al., 2023), which replaces softmax-based attention with a decaying matrix component. This allows Sable to model sequences flexibly: either as recurrent (RNN-like), parallel (attention-like), or chunkwise (a hybrid of both). During training, Sable uses chunkwise retention for efficient parallel computation and gradient flow, while execution relies on a recurrent mode that maintains a hidden state to capture temporal dependencies and ensure memory-efficient inference. Sable's architecture consists of an encoder that processes per-agent observations into compact observation embeddings and value estimates, and a decoder that outputs predicted logits and actions. Training is performed via standard policy gradient, using the PPO objective (Schulman et al., 2017). Further details on Sable's retention mechanism can be found in the Appendix.

**Implicit constraint Q-learning (ICQ) – effective offline regularisation.** The key idea in ICQ (Yang et al., 2021) is to avoid out-of-distribution actions in the offline setting by computing target Q-values sampled from the behaviour policy $\mu$ (extracted from the dataset), instead of $\pi$, such that the Bellman operator becomes $(\mathcal{T}^{\pi}Q)(\tau, a) = r + \gamma \mathbb{E}_{a' \sim \mu}[\rho(\tau', a')Q(\tau', a')]$, where $\rho(\tau', a') = \frac{\pi(a'|\tau')}{\mu(a'|\tau')}$ is an importance sampling weight. However, obtaining an accurate $\mu$ is itself difficult. Therfore, ICQ instead constrains policy updates such that $D_{KL}(\pi \parallel \mu)[\tau] \leq \epsilon$, with the corresponding optimal policy taking the form $\pi_{k+1}^*(a|\tau) = \frac{1}{Z(\tau)}\mu(a|\tau)\exp\left(\frac{Q_{\pi_k}(\tau, a)}{\alpha}\right)$. Here, $\alpha > 0$ is a Lagrangian coefficient in the unconstrained optimisation objective and $Z(\tau) = \sum_{\tilde{a}} \mu(\tilde{a}|\tau)\exp\left(\frac{1}{\alpha}Q_{\pi_k}(\tau, \tilde{a})\right)$ is the normalisation function. Finally, solving for $\rho(\tau', a') = \frac{\pi_{k+1}^*(a'|\tau')}{\mu(a'|\tau')}$, gives the ICQ operator (that only requires sampling from $\mu$) as

$$\mathcal{T}_{ICQ}Q(\tau, a) = r + \gamma \mathbb{E}_{a' \sim \mu}\left[\frac{1}{Z(\tau')}\exp\left(\frac{Q(\tau', a')}{\alpha}\right)Q(\tau', a')\right]. \quad (3)$$

During training, the critic network parameters $\phi$, and the policy network parameters $\theta$, are optimised over a data batch $\mathcal{B}$ by minimising the following losses:

**Critic loss** :

$$J_Q(\phi) = \mathbb{E}_{\tau, a, \tau', a' \sim \mathcal{B}}\left[r + \gamma \frac{1}{Z(\tau')}\exp\left(\frac{Q_{\phi^-}(\tau', a')}{\alpha}\right)Q_{\phi^-}(\tau', a') - Q_\phi(\tau, a)\right]^2$$

**Policy loss** :

$$J_\pi(\theta) = \mathbb{E}_{\tau \sim \mathcal{B}}\left[D_{KL}(\pi_{k+1}^* \parallel \pi_\theta)[\tau]\right] = \mathbb{E}_{\tau, a \sim \mathcal{D}}\left[-\frac{1}{Z(\tau)}\log(\pi_\theta(a|\tau))\exp\left(\frac{Q(\tau, a)}{\alpha}\right)\right]$$

Here, $\phi^-$ denotes the target network parameters. In practice, the normalising partition function is computed as $\sum_{(\tau, a) \in \mathcal{B}} \exp Q(\tau, a)/\alpha$, over the mini-batch $\mathcal{B}$ sampled from the dataset $\mathcal{D}$.

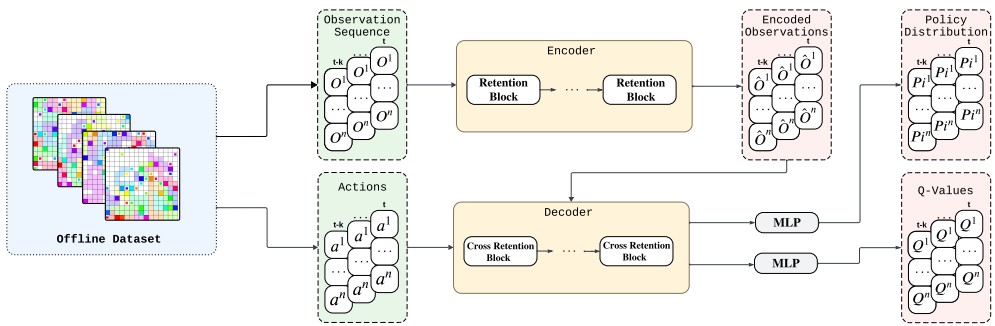

Figure 1: *Oryx's model architecture.* The green blocks indicate the inputs to the model (in yellow), sourced from the dataset of online experiences (in blue). First, a sequence of agent observations from timestep $t$ to $t+k$ is passed through the encoder. Inside each retention block, the network performs joint reasoning over the agents $(a_1, \ldots, a_n)$ and temporal context $(t, \ldots, t+k)$, producing encoded representations at each timestep. These encoded observations, along with the actions from the dataset, are passed to the decoder, which has two heads. One head returns Q-values, while the second returns a policy distribution for each agent for the full sequence.

## 3 Method

Oryx integrates the strengths of Sable's auto-regressive retention-based architecture and ICQ's robust offline regularisation to address the specific challenges of long-horizon miscoordination and accumulating extrapolation error in offline MARL. A particularly effective model class that can naturally instantiate autoregressive policies is sequence models, with notable examples the multi-agent transformer (MAT) (Wen et al., 2022) and Sable (Mahjoub et al., 2025). These models naturally represent the joint policy as a product of factors that similarly decompose auto-regressively as $\pi(\boldsymbol{a}|\boldsymbol{\tau}) = \prod_{j=1}^{n} \pi^{i_j}(a^{i_j}|\boldsymbol{\tau}, \mathbf{a}^{i_{1:j-1}})$.

**Network architecture.** Oryx modifies the Sable architecture, which features retention blocks to efficiently handle long sequential dependencies among agents. In particular, we employ a dual-output decoder structure where the decoder outputs both policy logits and Q-value estimates. The logits correspond to the action probabilities for each agent and the Q-value estimates explicitly capture the relative value of each agent's actions given the current historical context. Unlike the original Sable network, we do not have a value head as part of the encoder and instead train the encoder and decoder end-to-end with the combined critic and policy losses. A diagram of the computational flow of the Oryx architecture is provided in Figure 1.

**Autoregressive ICQ loss.** Next, we use the Oryx network's autoregressive policy structure and the advantage decomposition theorem (Kuba et al., 2021) to derive Theorem 1 (for proof see Appendix).

**Theorem 1.** *For an auto-regressive model, the multi-agent joint-policy under ICQ regularisation can be optimised sequentially for $j = 1, ..., n$ over a data batch $\mathcal{B}$ as follows:*

$$\pi_*^{i_j} = \underset{\pi^{i_j}}{\arg\max} \, \mathbb{E}_{\boldsymbol{\tau}, \mathbf{a}^{i_{1:j}} \sim \mathcal{B}} \left[ -\frac{1}{Z^{i_{1:j}}(\boldsymbol{\tau})} \log(\pi^{i_j}(a^{i_j} \mid \boldsymbol{\tau}, \mathbf{a}^{i_{1:j-1}})) \exp\left( \frac{A^{i_{1:j}}(\boldsymbol{\tau}, \mathbf{a}^{i_{1:j}})}{\alpha} \right) \right],$$

*where $Z^{i_{1:j}}(\boldsymbol{\tau}) = \prod_{l=1}^{j} \sum_{\tilde{a}^{i_l}} \mu^{i_l}(\tilde{a}^{i_l}|\boldsymbol{\tau}, \mathbf{a}^{i_{1:l-1}}) \exp\left( \frac{A^{i_{1:l}}(\boldsymbol{\tau}, \mathbf{a}^{i_{1:l}})}{\alpha} \right).$*

To arrive at a sequential SARSA-like algorithm similar to ICQ, we update the critic by sampling target actions from the dataset and update the Q-value function with implicit importance weights as

$$Q_{k+1}^{i_j} = \underset{Q^{i_j}}{\arg\min} \, \mathbb{E}_{\mathcal{B}} \left[ \left( \left( r + \gamma \frac{\exp\left( \frac{Q_{\phi^-}^{i_j}(\boldsymbol{\tau}', \mathbf{a}'^{,i_{1:j}})}{\alpha} \right)}{Z(\boldsymbol{\tau}')} Q_{\phi^-}^{i_j}(\boldsymbol{\tau}', \mathbf{a}'^{,i_{1:j}}) - Q_{\phi}^{i_j}(\boldsymbol{\tau}, \mathbf{a}^{i_{1:j}}) \right)^2 \right].$$

Finally, the centralised advantage estimate in the multi-agent policy gradient is susceptible to high variance. In particular, Kuba et al. (2021) provides an upper bound on the difference in gradient

variance between independent and centralised learning when using the standard V-value function as a baseline as $(n-1)\frac{(\epsilon B_i)^2}{1-\gamma^2}$, where $B_i = \sup_{\boldsymbol{\tau},\mathbf{a}} ||\nabla_{\theta^i} \log \pi_\theta^i(\hat{a}^i|\boldsymbol{\tau})||$, $\epsilon_i = \sup_{\boldsymbol{\tau},\mathbf{a}^{-i},a^i} |A^i(\boldsymbol{\tau}, \mathbf{a}^{-i}, a^i)|$, and $\epsilon = \max_i \epsilon_i$. This bound grows linearly in the number of agents. We can arrive at a better bound that removes the $(n-1)$ term by employing a counterfactual baseline (Kuba et al., 2021) as used in Foerster et al. (2018) such that

$$A^{i_{1:j}}(\boldsymbol{\tau}, \mathbf{a}^{i_{1:j}}) = \sum_{m=1}^{j} \left[ Q(\boldsymbol{\tau}, \mathbf{a}^{i_{1:m}}) - \sum_{a^{i_m}} \pi^{i_m}(a^{i_m} \mid \boldsymbol{\tau}, \mathbf{a}^{i_{1:m-1}}) Q(\boldsymbol{\tau}, \mathbf{a}^{i_{1:m}}) \right].$$

This completes the sequential updating scheme for Oryx. A full algorithm description is provided in Algorithm 1. Next, we describe the architectural details allowing Oryx to model long-term dependencies across sampled trajectories from the dataset.

---

**Algorithm 1** Oryx's sequential updating scheme with autoregressive ICQ regularisation

---

1: Initialise the joint policy and critic network parameters $\theta$, $\phi$.
2: **for** $k = 0, 1, \ldots$ **do**
3:     Sample a mini-batch $\mathcal{B} = \{(\boldsymbol{\tau}, \mathbf{a}^{i_{1:j}}, \boldsymbol{\tau}', \mathbf{a}'^{,i_{1:j}})\}$ of trajectories from the offline dataset $\mathcal{D}$.
4:     Draw a permutation $i_{1:n}$ of agents at random.
5:     **for** $j = 1 : n$ **do**
6:         Update the critic:

$$Q_{k+1}^{i_j} = \underset{Q^{i_j}}{\operatorname{argmin}} \mathbb{E}_{\mathcal{B}} \left[ \left( \left( r + \gamma \frac{\exp\left( \frac{Q_{\phi^-}^{i_j}(\boldsymbol{\tau}', \mathbf{a}'^{,i_{1:j}})}{\alpha} \right)}{Z(\boldsymbol{\tau}')} Q_{\phi^-}^{i_j}(\boldsymbol{\tau}', \mathbf{a}'^{,i_{1:j}}) - Q_\phi^{i_j}(\boldsymbol{\tau}, \mathbf{a}^{i_{1:j}}) \right) \right)^2 \right]$$

7:         Calculate the advantage:

$$A^{i_{1:j}}(\boldsymbol{\tau}, \mathbf{a}^{i_{1:j}}) = \sum_{m=1}^{j} \left[ Q(\boldsymbol{\tau}, \mathbf{a}^{i_{1:m}}) - \sum_{a^{i_m}} \pi^{i_m}(a^{i_m} \mid \boldsymbol{\tau}, \mathbf{a}^{i_{1:m-1}}) Q(\boldsymbol{\tau}, \mathbf{a}^{i_{1:m}}) \right]$$

8:         Update the policy:

$$\pi_{k+1}^{i_j} = \underset{\pi^{i_j}}{\operatorname{argmin}} \mathbb{E}_{\mathcal{B}} \left[ -\frac{1}{Z(\boldsymbol{\tau})} \log(\pi^{i_j}(a^{i_j} \mid \boldsymbol{\tau}, \mathbf{a}^{i_{1:j-1}})) \exp\left( \frac{A^{i_{1:j}}(\boldsymbol{\tau}, \mathbf{a}^{i_{1:j}})}{\alpha} \right) \right]$$

9:     **end for**
10: **end for**

---

# 4  Results

In this section, we conduct detailed empirical evaluations to substantiate the core contributions of our proposed algorithm, Oryx. We begin by rigorously validating its key design components: (i) sequential action selection for agent coordination, (ii) a memory mechanism for temporal coherence, and (iii) autoregressive ICQ for stable offline training. To assess scalability, we subject Oryx to a demanding multi-agent setting involving up to 50 agents. Specifically, we use the Connector environment (Bonnet et al., 2024), which progressively increases coordination complexity as the number of agents grows. Finally, we perform an extensive and diverse benchmarking study across more than 65 datasets from prominent MARL benchmarks, covering a wide range of tasks in SMAC (Samvelyan et al., 2019), MAMuJoCo (Peng et al., 2021), and RWARE (Papoudakis et al., 2021).

## 4.1  Validating Oryx's core mechanisms

Inspired by Osband et al. (2020), we designed a T-Maze environment (see Figure 2) to specifically isolate a long-horizon multi-agent coordination challenge. On the first timestep, two agents each independently select a colour ("green" or "orange"). On the subsequent timestep, they spawn randomly at the bottom of the maze and briefly observe their choice of goal action and the goal locations (e.g.,

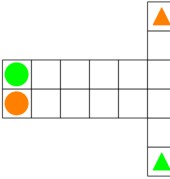

(a) T-Maze

| | | T-MAZE | |
|---|---|---|---|
| **Algorithm** | | Replay | Expert |
| **I-ICQ** | | $0.0 \pm 0.0$ | $0.0 \pm 0.0$ |
| **MAICQ** | | $0.0 \pm 0.0$ | $0.0 \pm 0.0$ |
| **Oryx - w/o Auto-Regressive Actions** | | $0.0 \pm 0.0$ | $0.0 \pm 0.0$ |
| **Oryx - w/o Memory** | | $0.58 \pm 0.04$ | $0.63 \pm 0.04$ |
| **Oryx - w/o ICQ** | | $0.0 \pm 0.0$ | $0.0 \pm 0.0$ |
| **Oryx** | | $\mathbf{0.99 \pm 0.01}$ | $\mathbf{0.94 \pm 0.03}$ |

(b) Performance across baselines and ablations

Figure 2: *Evaluating long horizon coordination.* To issolate the importance of the different components of Oryx a minimal two-agent environment, T-Maze was designed. In the environment the target states are revealed only at the first timestep, requiring agents to retain goal information throughout the episode and carefully coordinate at the end. **Oryx successfully solves the task only when all components are present, while baseline methods fail to perform across both the replay and expert datasets.**

orange-left, green-right). Without further observing their goal, agents must navigate to their respective targets, manoeuvring at the junction to avoid collision and delay. Success requires: (i) selecting different colours initially for effective coordination, (ii) retaining memory of their goal choice action, and (iii) efficiently manoeuvring around one another. Agents receive a team reward of 1 if both agents reach the goal. We generated two datasets: a `mixed` dataset containing primarily unsuccessful trajectories and a smaller number of successful examples, and an `expert` dataset consisting solely of successful trajectories. Dataset statistics are detailed in the appendix following Formanek et al. (2024a). We evaluated several baselines: i) fully independent ICQ learners (**I-ICQ**), ii) the CTDE variant with mixing networks (**MAICQ** (Yang et al., 2021)), and iii) targeted Oryx ablations disabling autoregressive action selection, memory, and offline sequential ICQ regularisation individually to isolate their contributions.

As shown in Figure 2, Oryx consistently achieves optimal performance with both datasets, while baseline ICQ variants fail regardless of value decomposition. The ablation results confirm that each of Oryx's core components—sequential action selection, memory, and ICQ-based offline regularisation—is individually crucial for enabling effective long-horizon coordination among agents.

## 4.2 Testing coordination in complex many-agent settings

Having validated the importance of Oryx's core components in a smaller-scale setting, we now stress-test the algorithm in significantly larger and more challenging many-agent coordination scenarios. To effectively evaluate this, we select the Connector environment from Bonnet et al. (2024), which inherently becomes more complex as agent density increases. It is important to highlight that naively increasing agent numbers does not necessarily enhance task complexity; it may even simplify certain problems where coordination is less critical. Despite its popularity, many SMAC scenarios tend to become easier as the number of agents increase (for example, reported performance often takes on the following task order, `5m_vs_6m` > `8m_vs_9m` > `10m_vs_11m` (Wen et al., 2022; Yu et al., 2022; Hu et al., 2023)). In Connector, agents spawn randomly on a fixed-size grid and are each assigned a random target location. Agents must navigate to their targets, leaving impassable trails behind them that can obstruct other agents. The necessity for careful coordination sharply increases as agent density grows, making Connector ideal for testing Oryx.

Datasets for Connector were generated by recording replay data from online training sessions using Sable (Mahjoub et al., 2025). Due to the substantial volume of data generated by the online systems (20M timesteps), we applied random uniform subsampling on each dataset to select 1M timesteps. Statistical summaries of these datasets are provided in the appendix. For comparison, we again use MAICQ (Yang et al., 2021) as our baseline. MAICQ is particularly valuable as a comparative algorithm because it incorporates several widely used components that differ notably from Oryx, specifically: (i) RNN-based memory, (ii) global state-conditioned value decomposition, and (iii) the original non-autoregressive ICQ loss. We normalise the results as in Fu et al. (2020)
$norm\_score = \frac{score - random\_score}{expert\_score - random\_score}$, where $random\_score$ is the average score achieved by

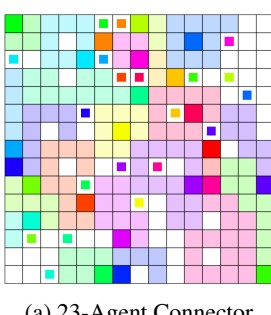

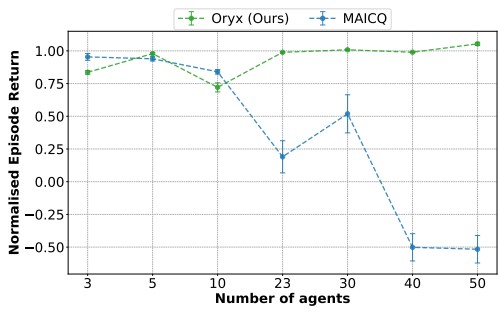

(a) 23-Agent Connector               (b) Results on Connector

Figure 3: *Evaluating Oryx on many-agent settings.* We compare Oryx, with its autoregressive ICQ loss and sequence model architecture, to MAICQ which is a non-autoregressive CTDE algorithm. The two algorithms are trained on datasets from Connector (Bonnet et al., 2024) scenarios with increasing numbers of agents. **While the performance of MAICQ dramatically degrades on scenarios with large numbers of agents, Oryx's performance remains robust.**

agents taking random actions and the $expert\_score$ is the performance of the online system at the end of training.

Our results indicate that at lower agent counts, performance differences between Oryx and MAICQ are modest. However, from around 23 agents, Oryx significantly outperforms MAICQ, achieving near-expert performance compared to only 25% of expert performance by MAICQ. Interestingly, between 23 and 30 agents, MAICQ's performance temporarily improves. However, this is likely due to the fact that we had to marginally increase the grid size at this point, making the environment slightly less dense (see Figure 7). Nonetheless, as agent count grows from 30 to 50 (with grid size held constant), coordination complexity dramatically escalates, clearly underscoring Oryx's superior capability to manage increasingly challenging coordination demands.

### 4.3 Demonstrating state-of-the-art performance on existing offline MARL datasets

Finally, we evaluate Oryx against a wide range of current state-of-the-art offline MARL algorithms across various widely recognised benchmark datasets. The datasets come from various original works, including Formanek et al. (2023a); Pan et al. (2022); Shao et al. (2023); Wang et al. (2023); Matsunaga et al. (2023) and were obtained via OG-MARL (Formanek et al., 2024a), a public datasets repository for offline MARL. We compare Oryx against the latest published performances on each dataset, including reported results from Matsunaga et al. (2023); Shao et al. (2023); Bui et al. (2025); Li et al. (2025). We follow a similar methodology to that of Formanek et al. (2024b), where we trained our algorithm for a fixed number of updates on each dataset and reported the mean episode return at the end of training over 320 rollouts. We repeated this procedure 10 times with different random seeds. We summarise our results in Figure 4 and provide detailed tabular data in the appendix, i.e. mean and standard deviations of episode returns across all random seeds for each dataset. We perform a simple heteroscedastic, two-sided t-test with a 95% confidence interval for testing statistical significance, following Papoudakis et al. (2021) and Formanek et al. (2024b).

**SMAC** is the most widely used environment in the offline MARL literature. However, as Formanek et al. (2024a) pointed out, different authors tend to not only use different datasets in their experiments but also entirely different scenarios. This makes comparing across works very challenging. As such, we tested Oryx across as many SMAC datasets that were available to us (Meng et al., 2022; Formanek et al., 2023a; Shao et al., 2023) and compared its performance against the latest state-of-the-art result we could find in the literature (Shao et al., 2023; Bui et al., 2025; Li et al., 2025). In total, we tested on 43 datasets, spanning 9 unique scenarios (2c_vs_64zg, 3s_vs_5z, 3m, 2s3z, 5m_vs_6m, 6h_vs_8z, 8m, 3s5z_vs_3s6z and corridor). **In total, Oryx matched or surpased the current state-of-the-art on 34 of the datasets (79%).**

**MAMuJoCo** is the most widely used multi-agent continuous control environment and second most popular source of datasets for testing offline MARL algorithms. We tested Oryx on datasets from Pan et al. (2022) and Wang et al. (2023). These span several scenarios with varying numbers of agents

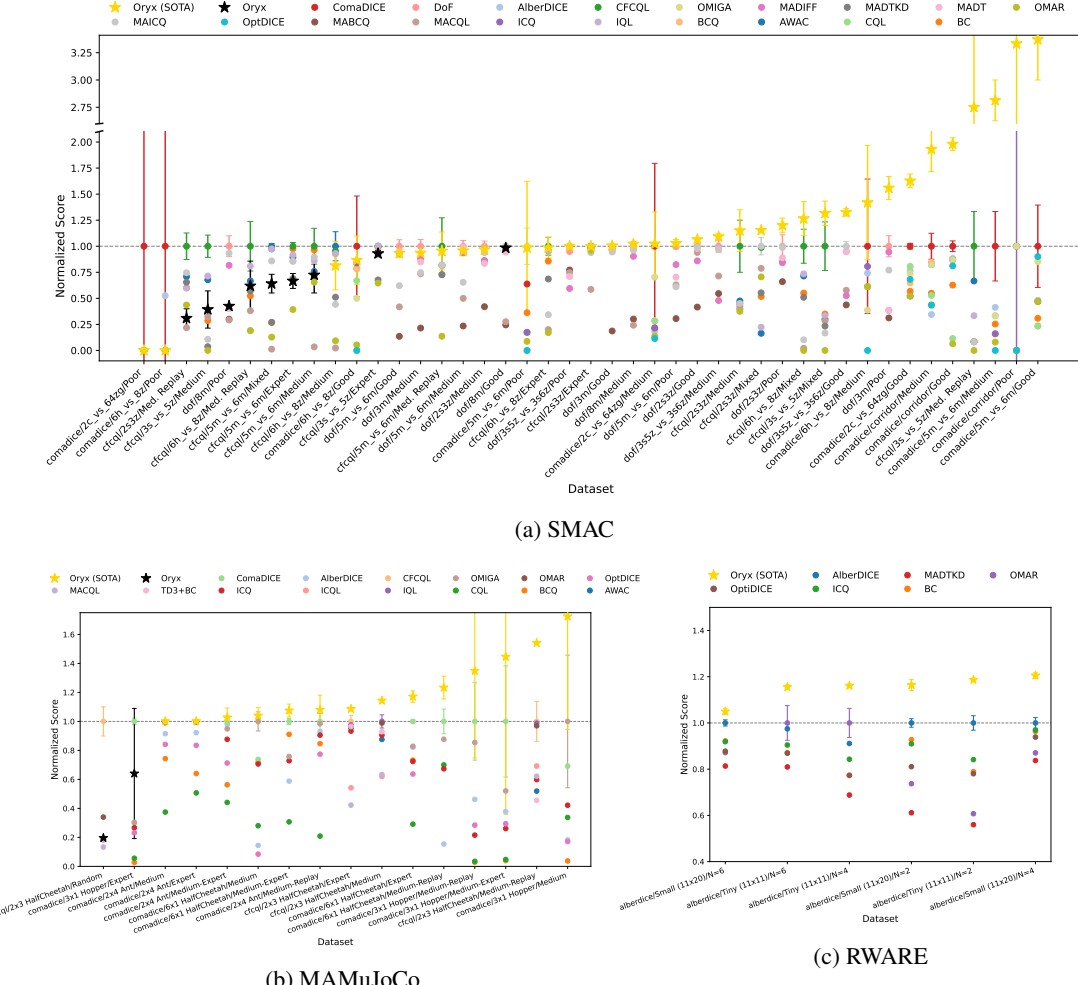

(a) SMAC

(b) MAMuJoCo

(c) RWARE

Figure 4: *Performance of Oryx across diverse benchmark datasets from prior literature.* Scores are normalised relative to the current state-of-the-art, with values above 1 indicating that Oryx surpasses previous best-known results. Unnormalized scores are provided in the appendix. **Gold stars indicate instances where Oryx matches or exceeds state-of-the-art performance, while black stars denote otherwise.**

including 2x3 HalfCheetah, 6x1 HalfCheetah, 2x4 Ant and 3x1 Hopper. We compared our results to recent state-of-the-art results by Shao et al. (2023) and Bui et al. (2025). **On 14 out of the 16 datasets tested, Oryx matched or surpassed the current state-of-the-art**.

**RWARE** (Papoudakis et al., 2021) is a well-known, challenging MARL environment that requires long-horizon coordination. Agents must learn to split up and avoid overlapping in order to optimally cover the warehouse. Moreover, episodes are quite long and have a very sparse reward signal, spanning 500 timesteps as compared to less than 100 and a dense reward on most SMAC scenarios. Matsunaga et al. (2023) generated and shared a number of datasets on RWARE, with varying numbers of agents ($n \in [2, 4, 6]$) and warehouse sizes. **Oryx set a new highest score on all six of the available datasets, and in several cases improved the state-of-the-art by nearly 20%.**

### 4.4 Comparing auto-regressive sequence modeling architectures

Oryx extends auregressive MARL sequence modeling into the offline MARL setting by leveraging Sable (Mahjoub et al., 2025) as its network architecture backbone. While prior sequence models such as MAT (Wen et al., 2022) have demonstrated strong performance in online settings, the decision

Table 1: *Comparing Oryx and MAT+ICQ.* In order to isolate the effect of using the Sable network architecture as the backbone of the Oryx network, we compare it to an offline varient of MAT that uses the same autoregressive ICQ loss as Oryx (MAT+ICQ). We report several aggregated metrics with 95% stratified bootstrap confidence intervals (Agarwal et al., 2021) across all SMAC and RWARE datasets. To Facilitate aggregation across scenarios with different episode returns, scores were first normalised by the highest return in each dataset respectivly. **We see from the results that on both environments the mean, median, interquartile mean (IQM), and optimality gap of Oryx is superior to MAT+ICQ**.

| | SMAC | | RWARE | |
|---|---|---|---|---|
| | **MAT+ICQ** | **Oryx** | **MAT+ICQ** | **Oryx** |
| Median ↑ | 0.71 [0.62, 0.74] | **0.91 [0.88, 0.92]** | 0.85 [0.84, 0.86] | **0.89 [0.88, 0.90]** |
| IQM ↑ | 0.67 [0.66, 0.69] | **0.87 [0.86, 0.88]** | 0.85 [0.84, 0.86] | **0.89 [0.89, 0.90]** |
| Mean ↑ | 0.63 [0.62, 0.64] | **0.77 [0.76, 0.79]** | 0.84 [0.83, 0.84] | **0.89 [0.88, 0.90]** |
| Optimality Gap ↓ | 0.38 [0.36, 0.39] | **0.23 [0.22, 0.25]** | 0.16 [0.16, 0.17] | **0.11 [0.10, 0.12]** |

to adopt Sable as the backbone for Oryx was motivated by its superior scalability, efficiency, and stability when trained across large populations of agents.

However, to further quantify the advantages of our design choices, we conducted an ablation comparing Oryx with an offline variant of MAT using identical training procedures and the same autoregressive ICQ loss. This experiment isolates the impact of choosing Sable over MAT for Oryx's sequence model component. Following the evaluation methodology proposed by Gorsane et al. (2022), we report aggregated metrics across all SMAC and RWARE datasets. To facilitate aggregation across scenarios with different expected episode returns, results were normalised by the highest episode return in each dataset. The results in Table 1 demonstrate that while MAT+ICQ is a strong baseline, Oryx still ourperforms it across all metrics and environments, validating our design decision to build on the Sable network architecture instead using MAT.

# 5 Related Work

Finally, we provide an overview of prior literature addressing the key challenges of offline MARL and works that approach MARL as a sequence modeling problem, highlighting connections and distinctions relative to our contributions.

**Offline MARL.** Early works such as Jiang and Lu (2021) and Yang et al. (2021) introduced methods to mitigate extrapolation errors through constrained Q-value estimation within the training distribution. Pan et al. (2022) combined first-order gradient methods with zeroth-order optimisation to guide policies toward high-value actions, with further refinements provided by Shao et al. (2023), which applied conservative regularisation using per-agent counterfactual reasoning, and Wang et al. (2023), employing a global-to-local value decomposition approach. Distributional constraints have been another promising direction, notably in works like Matsunaga et al. (2023) and Bui et al. (2025). Numerous additional works have also tackled extrapolation error, coordination issues, and offline stability (Zhang et al., 2023a; Wu et al., 2023; Eldeeb et al., 2024; Liu et al., 2024b; Barde et al., 2024; Liu et al., 2025; Zhou et al., 2025). Our approach builds directly upon these insights, specifically extending the early ICQ framework (Yang et al., 2021) to enhance learning stability from offline trajectories.

Moreover, theoretical advancements have enhanced understanding of the guarantees and limitations inherent to offline MARL (Cui and Du, 2022b,a; Zhong et al., 2022; Zhang et al., 2023b; Xiong et al., 2023; Wu et al., 2023). Complementary studies investigated opponent modeling in offline contexts (Jing et al., 2024), and explorations into offline-to-online transitions (Zhong et al., 2024a; Formanek et al., 2023b) reveal potential pathways to bridge static offline training with dynamic online adaptation.

**MARL as a Sequence Modeling Problem.** Capturing long-term dependencies and joint-agent behaviour is critical, particularly in partially observable scenarios and tasks with extended horizons. Prior works such as Meng et al. (2022) and Tseng et al. (2022) addressed these challenges through Transformer-based architectures, effectively modeling trajectory data in offline MARL settings.

Attention-based and diffusion-based strategies have also been explored (Zhu et al., 2024; Qiao et al., 2025; Li et al., 2025; Fu et al., 2025). Online MARL research further contributed with influential works like by Wen et al. (2022), which introduced a transformer-based autoregressive action selection mechanism, alongside newer architectures aimed at better scalability with number of agents (Daniel et al., 2025; Mahjoub et al., 2025).

# 6    Conclusion

In this work, we introduced Oryx, a novel algorithm explicitly designed to address the critical challenges of coordination amongst many agents in offline MARL. We derived a sequential policy updating scheme that leverages implicit constraint Q-learning (Yang et al., 2021) and the advantage decomposition theorem (Kuba et al., 2021). By integrating this sequential ICQ-based updating scheme with a modified version of the Sable network (Mahjoub et al., 2025), Oryx effectively mitigates two fundamental problems in offline MARL: extrapolation error and miscoordination. Our extensive empirical evaluation across diverse benchmarks—including SMAC, RWARE, Multi-Agent MuJoCo, and Connector—demonstrates that Oryx consistently achieves state-of-the-art results. Notably, Oryx excels in scenarios characterised by high agent densities and complex coordination requirements, distinguishing it from prior approaches for offline multi-agent learning. To help accelerate research in this direction, we make all our datasets on Connector, with up to 50 agents, openly accessible to the community.

**Limitations and future work**    While Oryx demonstrates robust performance across diverse research benchmarks, its evaluation is naturally limited compared to the true complexity of large-scale, real-world industrial settings. Consequently, important future work includes extending Oryx to hybrid offline-online settings and evaluating its effectiveness in broader, more varied domains, particularly real-world applications. Furthermore, our work introduces autoregressive policies to offline MARL, demonstrating the significant promise of utlising such policies in other architectures and setups. Future research could extend existing and novel offline MARL methods to utilise autoregressive policies (Fu et al., 2022) and sequence models. Integrating diverse offline learning techniques into this paradigm may lead to the discovery of even more effective sequence models for multi-agent coordination.

# Acknowledgments

We would like to thank Louise Beyers for her valuable contributions at the outset of this project, which were crucial for its inception. We thank our MLOps team for developing our model training and experiment orchestration platform AIchor. We thank the Python and JAX communities for developing tools that made this research possible. We thank the anonymous reviewers for their constructive feedback and valuable suggestions. Finally, we thank Google's TPU Research Cloud (TRC) for supporting our research with Cloud TPUs.

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

# A Additional Algorithm Details

## A.1 Proof of Theorem 1

**Theorem 1.** *For an auto-regressive model, the multi-agent joint-policy under ICQ regularisation can be optimised for $j = 1, ..., n$ as follows:*

$$\pi_*^{i_j} = \underset{\pi^{i_j}}{\operatorname{argmax}} \, \mathbb{E}_{\boldsymbol{\tau}, \mathbf{a}^{i_{1:j}} \sim \mathcal{B}} \left[ -\frac{1}{Z^{i_{1:j}}(\boldsymbol{\tau})} \log(\pi^{i_j}(a^{i_j} \mid \boldsymbol{\tau}, \mathbf{a}^{i_{1:j-1}})) \exp \left( \frac{A^{i_{1:j}}(\boldsymbol{\tau}, \mathbf{a}^{i_{1:j}})}{\alpha} \right) \right],$$

*where $Z^{i_{1:j}}(\boldsymbol{\tau}) = \prod_{l=1}^{j} \sum_{\tilde{a}^{i_l}} \mu^{i_l}(\tilde{a}^{i_l} | \boldsymbol{\tau}, \mathbf{a}^{i_{1:l-1}}) \exp \left( \frac{A^{i_{1:l}}(\boldsymbol{\tau}, \mathbf{a}^{i_{1:l}})}{\alpha} \right)$ is the normalisation function.*

*Proof.* For an auto-regressive model, we can decompose the joint policy as

$$\pi(\boldsymbol{a}|\boldsymbol{\tau}) = \pi^{i_{1:n}}(\boldsymbol{a}^{i_{1:n}}|\boldsymbol{\tau}) = \prod_{j=1}^{n} \pi^{i_j}(a^{i_j}|\boldsymbol{\tau}, \mathbf{a}^{i_{1:j-1}}).$$

Furthermore, from the advantage decomposition theorem by Kuba et al. (2021), we have that

$$A(\boldsymbol{\tau}, \mathbf{a}) = A^{i_{1:n}}(\boldsymbol{\tau}, \mathbf{a}^{i_{1:n}}) = \sum_{j=1}^{n} A^{i_j}(\boldsymbol{\tau}, \mathbf{a}^{i_{1:j-1}}, a^{i_j}).$$

Let $\mathcal{J}_{\boldsymbol{\pi}}$ denote the joint-policy loss as in Yang et al. (2021). Using the above decompositions we can re-write the loss as follows

$$\mathcal{J}_{\boldsymbol{\pi}} = \mathbb{E}_{\boldsymbol{\tau}, \mathbf{a} \sim B} \left[ -\frac{1}{Z(\boldsymbol{\tau})} \log(\pi(\mathbf{a}|\boldsymbol{\tau})) \exp \left( \frac{A(\boldsymbol{\tau}, \mathbf{a})}{\alpha} \right) \right]$$

$$= \mathbb{E}_{\boldsymbol{\tau}, \mathbf{a}^{i_{1:n}} \sim B} \left[ -\frac{1}{Z(\boldsymbol{\tau})} \left( \sum_{j=1}^{n} \log(\pi^{i_j}(a^{i_j} \mid \boldsymbol{\tau}, \mathbf{a}^{i_{1:j-1}})) \right) \exp \left( \frac{\sum_{l=1}^{n} A^{i_l}(\boldsymbol{\tau}, \mathbf{a}^{i_{1:l-1}}, a^{i_l})}{\alpha} \right) \right]$$

$$= \sum_{j=1}^{n} \mathbb{E}_{\boldsymbol{\tau}, \mathbf{a}^{i_{1:n}} \sim B} \left[ -\frac{1}{Z(\boldsymbol{\tau})} \left( \log(\pi^{i_j}(a^{i_j} \mid \boldsymbol{\tau}, \mathbf{a}^{i_{1:j-1}})) \right) \exp \left( \frac{\sum_{l=1}^{n} A^{i_l}(\boldsymbol{\tau}, \mathbf{a}^{i_{1:l-1}}, a^{i_l})}{\alpha} \right) \right]$$

$$= \sum_{j=1}^{n} \mathbb{E}_{\boldsymbol{\tau}, \mathbf{a}^{i_{1:j}}, \mathbf{a}^{i_{j+1:n}} \sim B} \left[ -\frac{1}{Z(\boldsymbol{\tau})} \log(\pi^{i_j}(a^{i_j} \mid \boldsymbol{\tau}, \mathbf{a}^{i_{1:j-1}})) \exp \left( \frac{\sum_{l=1}^{j} A^{i_l}(\boldsymbol{\tau}, \mathbf{a}^{i_{1:l-1}}, a^{i_l})}{\alpha} \right) \right.$$

$$\left. \cdot \exp \left( \frac{\sum_{k=j+1}^{n} A^{i_k}(\boldsymbol{\tau}, \mathbf{a}^{i_{1:k-1}}, a^{i_k})}{\alpha} \right) \right]$$

$$= \sum_{j=1}^{n} \mathbb{E}_{\boldsymbol{\tau}, \mathbf{a}^{i_{1:j}} \sim B} \left[ -\frac{1}{Z^{i_{1:j}}(\boldsymbol{\tau})} \log(\pi^{i_j}(a^{i_j} \mid \boldsymbol{\tau}, \mathbf{a}^{i_{1:j-1}})) \exp \left( \frac{\sum_{l=1}^{j} A^{i_l}(\boldsymbol{\tau}, \mathbf{a}^{i_{1:l-1}}, a^{i_l})}{\alpha} \right) \right]$$

$$\cdot \mathbb{E}_{\boldsymbol{\tau}, \mathbf{a}^{i_{1:n}} \sim B} \left[ \frac{1}{Z^{i_{j+1:n}}(\boldsymbol{\tau})} \exp \left( \frac{\sum_{k=j+1}^{n} A^{i_k}(\boldsymbol{\tau}, \mathbf{a}^{i_{j+1:k-1}}, a^{i_k})}{\alpha} \right) \right]$$

$$= \sum_{j=1}^{n} \mathbb{E}_{\boldsymbol{\tau}, \mathbf{a}^{i_{1:j}} \sim B} \left[ -\frac{1}{Z^{i_{1:j}}(\boldsymbol{\tau})} \log(\pi^{i_j}(a^{i_j} \mid \boldsymbol{\tau}, \mathbf{a}^{i_{1:j-1}})) \exp \left( \frac{A^{i_{1:j}}(\boldsymbol{\tau}, \mathbf{a}^{i_{1:j}})}{\alpha} \right) \right]$$

where the normalising partition function is given as

$$Z(\boldsymbol{\tau}) = \prod_{m=1}^{n} \sum_{\tilde{a}^{i_m}} \mu^{i_m}(\tilde{a}^{i_m}|\boldsymbol{\tau}, \mathbf{a}^{i_{1:m-1}}) \exp \left( \frac{A^{i_{1:m}}(\boldsymbol{\tau}, \mathbf{a}^{i_{1:m}})}{\alpha} \right),$$

and for any specific $j = 1, ..., n$ can be factorised as

$$Z(\boldsymbol{\tau}) = \prod_{l=1}^{j} \sum_{\tilde{a}^{i_l}} \mu^{i_l}(\tilde{a}^{i_l}|\boldsymbol{\tau}, \mathbf{a}^{i_{1:l-1}}) \exp \left( \frac{A^{i_{1:l}}(\boldsymbol{\tau}, \mathbf{a}^{i_{1:l}})}{\alpha} \right) \cdot \prod_{k=j+1}^{n} \sum_{\tilde{a}^{i_k}} \mu^{i_k}(\tilde{a}^{i_k}|\boldsymbol{\tau}, \mathbf{a}^{i_{1:k-1}}) \exp \left( \frac{A^{i_{1:k}}(\boldsymbol{\tau}, \mathbf{a}^{i_{1:k}})}{\alpha} \right)$$

$$= Z^{i_{1:j}}(\boldsymbol{\tau}) \cdot Z^{i_{j+1:n}}(\boldsymbol{\tau})$$

$\square$

## A.2 About Sable

Sable employs a Retention-based architecture designed to efficiently model long-range temporal dependencies amongst many agents. The model operates in two distinct modes depending on the phase: *recurrent execution* and *chunkwise training*.

**Execution Phase.** During interaction with the environment, Sable runs in *recurrent mode*, maintaining a retention state that evolves over time. This mode supports memory- and compute-efficient inference, as it scales linearly with the number of agents and is constant with respect to time.

**Training Phase.** For gradient-based optimisation, Sable leverages *chunkwise mode*, a parallel variant of Retention that processes fixed-length temporal chunks. Unlike fully parallel mechanisms, the chunkwise representation preserves hidden state transitions across chunk boundaries. This design allows the model to propagate temporal signals through time respecting episode boundaries (resettable retention). The choice of chunkwise training over parallel mode ensures that the retention state $\{S_\tau\}$ can be passed between training chunks $\tau$, maintaining temporal coherence and improving convergence stability.

**Oryx Adaptation.** In our offline variant, Oryx, we adopt only the *chunkwise* training mode from Sable. Since training occurs entirely off-policy datasets (offline), we do not maintain a persistent recurrent retention state $\{S_\tau\}$. Instead, whenever we initiate the training phase, we begin with a `None` hidden state, ensuring no temporal leakage from prior training phases.

As mentioned in the main text, our adaptation of Sable to Oryx includes additional structural modifications to better support the ICQ loss. Specifically, we remove the value head from the encoder and use it solely to produce observation representations. The decoder, in turn, is extended to output both the action distribution and Q-values, enabling compatibility with the ICQ training objective.

# B   T-MAZE

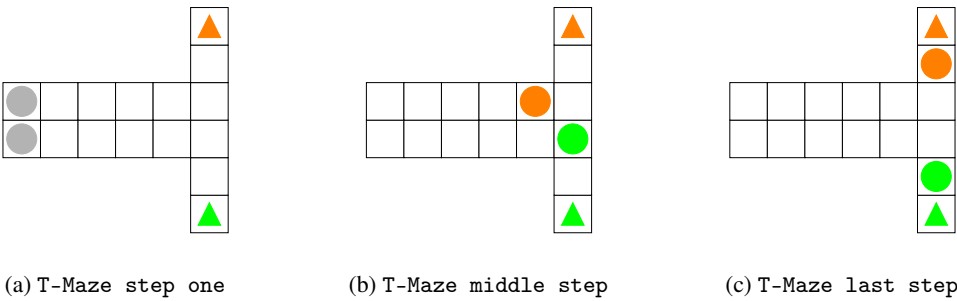

(a) `T-Maze step one`   (b) `T-Maze middle step`   (c) `T-Maze last step`

Figure 5: Environment visualisation for Connector.

## B.1   Environment Details

The T-Maze environment is intentionally designed as a minimalist setting that isolates key challenges for multi-agent reinforcement learning: interdependent action selection, reliance on memory from previous timesteps, and effective coordination to achieve a common goal. The environment unfolds in two distinct phases.

**Phase 1: Initial Target Color Selection**

This phase spans a single timestep at the beginning of each episode and is visualized in Figure Figure 5 (a). During this step:

- **Observation:** Agents receive no specific environmental observation.
- **Action Space:** Each agent must independently choose one of two actions: *choose orange* or *choose green*.
- **Coordination Requirement:** For the episode to be solvable, the two agents must select different target colors. If both agents choose the same color, they will be assigned the same goal location, making successful completion impossible. This necessitates a coordinated action selection strategy at the outset.

**Phase 2: Navigation and Goal Achievement** This phase encompasses all subsequent timesteps until the episode terminates. It is visualised in Figure Figure 5 (b) and (c)

- Initial Setup (Start of Phase 2):
  - **Agent Placement:** The two agents are randomly assigned to one of two distinct starting squares located at the base of the T-maze stem.
  - **Goal Assignment:** The two target locations (at the ends of the T-maze arms) are randomly assigned the colors green and orange, ensuring one goal is green and the other is orange.
- **Observation Space (During Navigation):** In each step of this phase, agents receive the following information:
  - A 3x3 local grid view, centered on the agent's current position, showing the maze structure and potentially the other agent if within this vicinity.
  - Information indicating which corridor contains the green target. From this, the location of the orange target in the opposite corridor can be inferred.
  - Their own action taken in the immediately preceding timestep.
- **Action Space (During Navigation):** Agents can choose to move in any of the four cardinal directions (up, down, left, right) or to take a *do nothing* action.
- **Memory Requirement:** The critical memory challenge arises from the observation structure. On the first timestep of Phase 2 (the second timestep of the episode overall), an agent observes the outcome of its *choose target* action from Phase 1 (i.e., it knows its chosen color

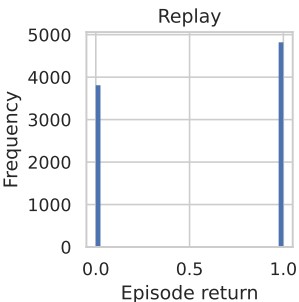

Figure 6: Distribution of episode returns of the recorded Replay Data for T-Maze.

and the corresponding goal location). For all subsequent timesteps in Phase 2, the agent only observes its more recent movement actions as its "previous action." Therefore, each agent must retain the memory of its initially chosen target color for the remainder of the episode to navigate correctly.

**Dynamics and Rewards**

- **Movement and Collision:** If an agent attempts to move into a wall or into a square occupied by the other agent, its position remains unchanged for that timestep.

- **Reward Structure:** Agents receive a sparse team reward. A reward of +1 is given if both agents are simultaneously positioned on their correctly colored target locations. In all other timesteps, and for all other outcomes (including collisions or incorrect goal locations), the reward is 0.

This design forces agents to first coordinate on distinct objectives, then remember their individual objective over a potentially long horizon while navigating a shared space and avoiding interference at junctions.

## B.2 Dataset Generation

For the T-Maze experiment, we generate two types of offline datasets: **replay** and **expert**. Both are collected from training Sable for 20 million timesteps. The replay dataset is recorded continuously during training by logging trajectories sampled throughout the execution phase. In contrast, the expert dataset is generated post-training by evaluating the final policy parameters for a fixed number of steps, producing trajectories that reflect near-expert behaviour.

The replay dataset contains over 16 million transitions with a mean episode return of 0.559, reflecting a mix of behaviours observed throughout training. While, the good dataset comprises 100,000 transitions collected by evaluating the final policy, with all episodes achieving a perfect return of 1.

## B.3 Algorithms Details

### B.3.1 Implementation Details

For baselines, we select MAICQ (Yang et al., 2021) and its fully independent variant (I-ICQ), in which we remove the QMixer component to enable decentralised training across agents.
Besides the baselines and Oryx, we test the effect of isolating Oryx's key components. The first ablation concerns *disabling auto-regressive actions*, in the original implementation, agent $i + 1$ receives the actions of agents $i, i-1, ..., 1$ as an input, within the same timestep, to support sequential coordination. To disable this mechanism, we instead feed a constant placeholder value of -1 as the previous action input to all agents during both training and evaluation. The second ablation targets *removing temporal dependency*, to achieve this, we reduce the sequence length from the default value of 20 (used in Oryx) to 2, such that only timesteps $t$ and $t + 1$ are provided during training. This limited context prevents the model from capturing or leveraging long-term temporal dependencies, effectively disabling its ability to retain memory across steps. The third ablation is about *removing*

*the ICQ loss component*, specifically, we eliminate the advantage estimation and policy regression terms from the loss function, retaining only standard Q-learning.

### B.3.2 Evaluation Details and Hyperparameters

Each offline system is trained for 100,000 gradient updates. Final performance is evaluated over 32 parallel episodes, with results aggregated across 10 independent training runs using different random seeds.

Table 2: Default hyperparameters for MAICQ and I-ICQ

| Parameter | Value |
|---|---|
| Sample sequence length | 20 |
| Sample batch size | 64 |
| Learning rate | 3e-4 |
| ICQ Value temperature | 1000 |
| ICQ Policy temperature | 0.1 |
| Linear layer dimension | 64 |
| Recurrent layer dimension | 64 |
| Mixer embedding dimension | 32 |
| Mixer hypernetwork dimension | 64 |

Table 3: Default hyperparameters for Oryx and its ablation variants

| Parameter | Value |
|---|---|
| Sample sequence length | 20 |
| Sample batch size | 64 |
| Learning rate | 3e-4 |
| ICQ Value temperature | 1000 |
| ICQ Policy temperature | 0.1 |
| Model embedding dimension | 64 |
| Number retention heads | 1 |
| Number retention blocks | 1 |
| Retention heads $\kappa$ scaling | 0.5 |

## C   Connector

### C.1   Environment Details

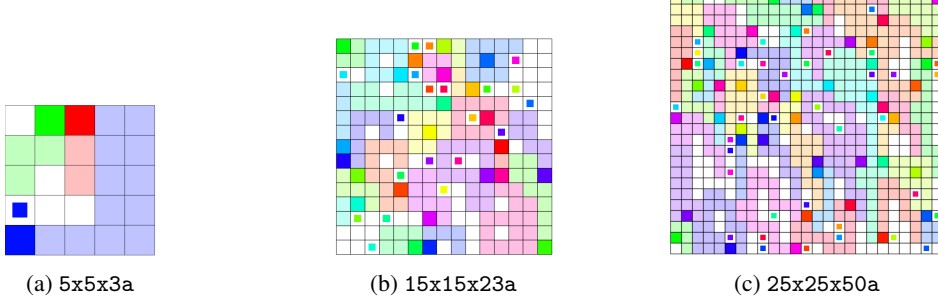

| (a) 5x5x3a | (b) 15x15x23a | (c) 25x25x50a |
|---|---|---|

Figure 7: Environment visualisation for Connector.

The Connector environment (Bonnet et al., 2024) is a cooperative multi-agent grid world where each agent is randomly assigned a start and end position and must construct a path to connect the two. As agents move, they leave behind impassable trails, introducing the need for coordination to avoid blocking others. The action space is discrete with five options: up, down, left, right, and no-op. Agents observe a local $d \times d$ view centered on their position, including visible trails, their own coordinates, and all target locations. The reward function assigns $+1$ when an agent successfully connects to its target and $-0.03$ at every other timestep, with no further reward once connected.

For scenario naming, we adopt the convention con-<x_size>x<y_size>-<num_agents>a, where each task is defined by the grid dimensions and the number of agents. In our experiments, we use the original Connector scenarios from the Sable paper (Mahjoub et al., 2025): 5x5x3a, 7x7x5a, 10x10x10a, and 15x15x23a. To evaluate scalability beyond 23 agents, we introduce three new scenarios—18x18x30a, 22x22x40a, and 25x25x50a—while approximately preserving agent density across scenarios.

### C.2   Dataset Details

To generate the offline datasets, we train Sable using the hyperparameters reported in the original Sable paper for each Connector task; for scenarios with more than 30 agents, we reuse the parameters from the 23-agent setting. For all tasks involving 3 to 30 agents, we train Sable for 20M timesteps and record the full execution data, resulting in 20M samples per task. We then uniformly subsample 1M transitions using the tools introduced in Formanek et al. (2024a), which randomly shuffle the set of available episodes and iteratively select full episodes until the target number of transitions is reached. For the larger 40-agent and 50-agent scenarios, due to memory constraints, we directly record less than 1M transitions during training and use them as-is without additional subsampling.

Table 4: Summary of Recorded Offline Datasets

| Task | Samples | Mean Return | Max Return | Min Return |
|---|---|---|---|---|
| con-5x5x3a | 1.17M | 0.59 | 0.97 | -0.75 |
| con-7x7x5a | 1.13M | 0.48 | 0.97 | -1.23 |
| con-10x10x10a | 1.09M | 0.40 | 0.97 | -1.53 |
| con-15x15x23a | 1.06M | 0.34 | 0.97 | -1.56 |
| con-18x18x30a | 1.00M | 0.25 | 0.97 | -2.43 |
| con-22x22x40a | 624,640 | 0.4 | 0.97 | -2.61 |
| con-25x25x50a | 624,640 | 0.33 | 0.97 | -3.06 |

### C.3   Extra Results on Default Connector

We evaluate Oryx against several well-established baselines on the Connector tasks with $n = 3, 5, 10, 23$: (i) MAICQ, along with its decentralised variant I-ICQ, (iii) Multi-Agent Decision

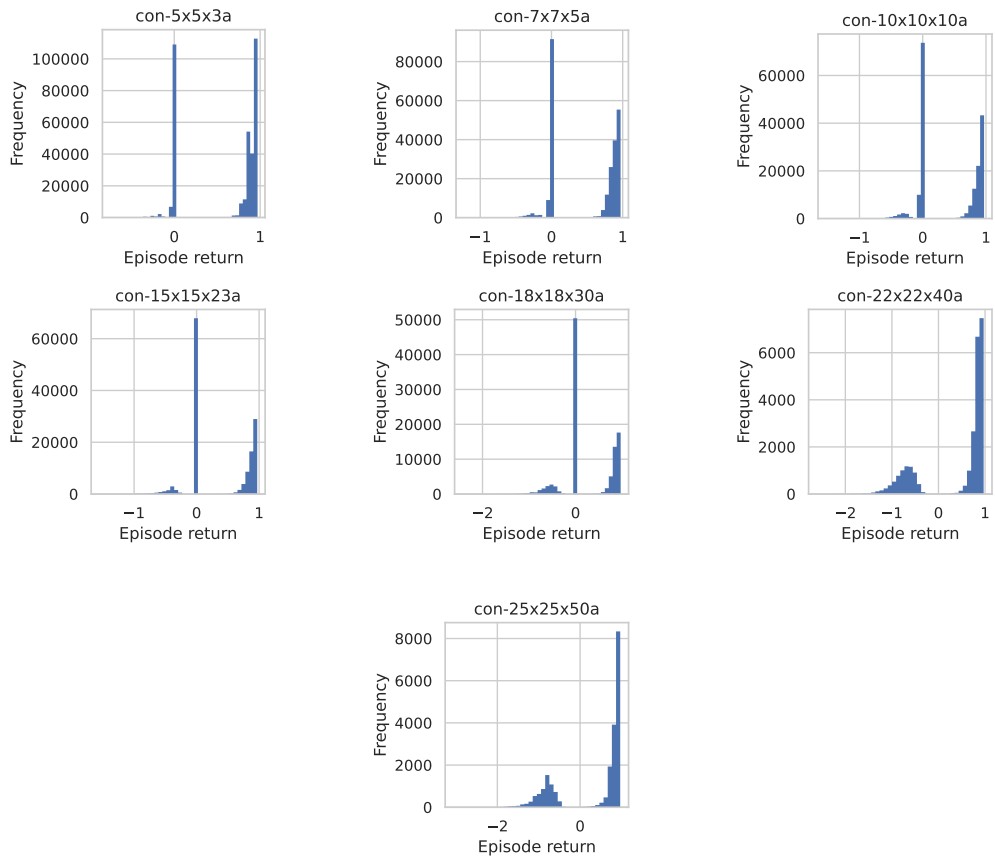

Figure 8: Distribution of episode returns of the recorded Replay Data for Connector.

Transformer (MADT) ([Meng et al., 2022](#)), and (iii) IQL-CQL. Both MAICQ and IQL-CQL are the implementations from OG-MARL ([Formanek et al., 2024b](#)). Based on their relative performance, we select the most competitive baseline for the scaling experiments ($n = 30, 40, 50$) reported in [subsection 4.2](#).

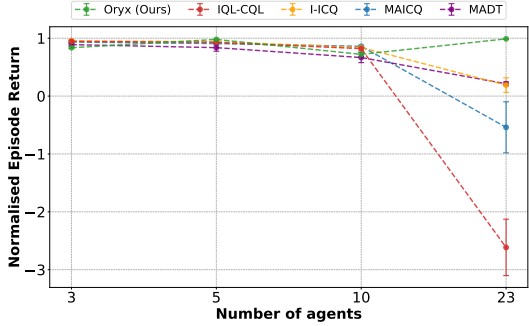

Figure 9: *Performance on the smaller Connector scenarios*—While all methods perform similarly in low-agent settings, Oryx begins to outperform the other baselines as the number of agents increases. I-ICQ and MADT follow. However, despite our best efforts, we could not run MADT on the larger agent instances without running out of Memory on our compute infrastructure. As such, I-ICQ offered a better compute-performance trade-off, making it the preferred baseline for scaling beyond 23 agents.

### C.4 Evaluation Details and Hyperparameters

For evaluation, we train Oryx and MAICQ for 100k gradient updates across 10 seeds (reduced to 3 seeds for settings with 40 and 50 agents due to computational constraints). Final performance is reported based on evaluation over 320 episodes.
We conducted preliminary comparisons between I-ICQ and MAICQ and observed that I-ICQ consistently outperformed the centralised training system (see Figure 9).

All the Connector experiments were distributed across NVIDIA A100 GPUs (40GB and 80GB VRAM).

Table 5: Default hyperparameters for MAICQ and I-ICQ

| Parameter | Value |
|---|---|
| Sample sequence length | 20 |
| Sample batch size | 64 |
| Learning rate | 3e-4 |
| ICQ Value temperature | 1000 |
| ICQ Policy temperature | 0.1 |
| Linear layer dimension | 128 |
| Recurrent layer dimension | 64 |
| Mixer embedding dimension (MAICQ only) | 32 |
| Mixer hypernetwork dimension (MAICQ only) | 64 |

Table 6: Default hyperparameters for IQL-CQL

| Parameter | Value |
|---|---|
| Sample sequence length | 20 |
| Sample batch size | 64 |
| Learning rate | 3e-4 |
| Linear layer dimension | 128 |
| Recurrent layer dimension | 64 |
| CQL weight | 3.0 |

Table 7: Default hyperparameters for Oryx

| Parameter | Value |
|---|---|
| Sample sequence length | 20 |
| Sample batch size | 64 |
| Learning rate | 3e-4 |
| ICQ Value temperature | 1000 |
| ICQ Policy temperature | 0.1 |
| Model embedding dimension | 128 |
| Number retention heads | 4 |
| Number retention blocks | 1 |
| Retention heads $\kappa$ scaling parameter | 0.5 |

For MADT, we adopt the default hyperparameters from the official repository of the original paper. However, we evaluated two configurations: (i) the default MADT setup, and (ii) an enhanced variant that includes reward-to-go, state, and action in the centralised observation. We found the latter consistently outperformed the default, and therefore use it in the reported results.

# D    SMAX

Before trying scaling experiments on Connector, we first evaluated Oryx on SMAX, an adaptation of SMAC(v2)(Ellis et al., 2023) implemented in JAX via JaxMARL(Rutherford et al., 2024), to assess its performance on a widely adopted benchmark in MARL.

## D.1    About SMAX

SMAX (Rutherford et al., 2024) is a JAX-based reimplementation of SMAC (Samvelyan et al., 2019) and SMAC(v2) (Ellis et al., 2023), designed for efficient experimentation without the need for the StarCraft II engine. In this environment, agents form teams of heterogeneous units and cooperate to win battles in a real-time strategy setting. Each agent observes local information—such as positions, health, unit types, and recent actions of nearby allies and enemies—and selects from a discrete action space including movement and attack commands. Unlike SMAC, SMAX balances reward signals between tactical engagements (damage dealt) and final success (winning the episode).

For the scaling experiments, we focus on SMAC(v2), evaluating performance across four scenarios of increasing agent count: `smacv2_5units`, `smacv2_10units`, `smacv2_20units`, and `smacv2_30units`.

## D.2    Scaling Agents in SMAX

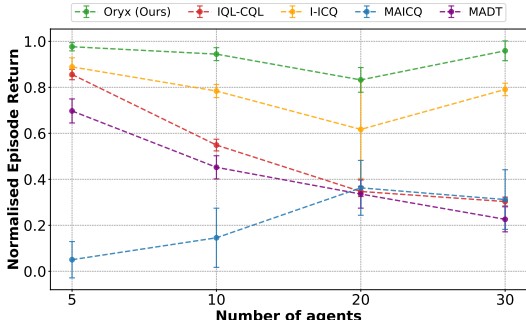

Figure 10: *Performance on SMAC(v2) across varying agent populations (5 to 30 agents)—* While SMAC scenarios do not always become harder with scale—some higher-agent tasks may in fact be less coordination-intensive—Oryx remains robust and achieves superior results across the board.

## D.3    Evaluation Details and Hyperparameters

For data collection, we follow the same protocol used for Connector (3–30 agents) as described in Section C.2. The evaluation procedure and the chosen hyperparameters are also consistent with those applied in the Connector experiments.

# E   Tabulated Benchmarking Results

Here we provide the tabulated results from our benchmark on SMAC, RWARE and MAMuJoCo. The algorithm with the highest mean episode return is denoted in bold. Algorithm that are not significantly different to the best, based on the two-sided t-test we conducted (Papoudakis et al., 2021; Formanek et al., 2024b), are denoted with an asterisk. All Oryx results are reported over 10 random seeds, with the mean and standard deviation given.

Table 8: Results (Win Rate) on SMAC datasets from OMIGA (Wang et al., 2023). Other algorithm results from ComaDICE (Bui et al., 2025).

| | 2c_vs_64zg | | | 5m_vs_6m | | | 6h_vs_8z | | | corridor | | |
|---|---|---|---|---|---|---|---|---|---|---|---|---|
| Algorithm | Good | Medium | Poor | Good | Medium | Poor | Good | Medium | Poor | Good | Medium | Poor |
| BC | 31.2 ± 9.9 | 1.9 ± 1.5 | 0.0 ± 0.0* | 2.5 ± 2.3 | 1.9 ± 1.5 | 2.5 ± 1.3 | 8.8 ± 1.2* | 1.9 ± 1.5 | 0.0 ± 0.0* | 30.6 ± 4.1 | 15.0 ± 2.3 | 0.0 ± 0.0 |
| BCQ | 35.6 ± 8.8 | 2.5 ± 3.6 | 0.0 ± 0.0* | 1.9 ± 2.5 | 1.2 ± 1.5 | 1.2 ± 1.5 | 8.8 ± 3.6* | 1.9 ± 1.5 | 0.0 ± 0.0* | 42.5 ± 6.4 | 23.1 ± 1.5 | 0.0 ± 0.0 |
| CQL | 44.4 ± 13.0 | 2.5 ± 3.6 | 0.0 ± 0.0* | 1.9 ± 1.5 | 2.5 ± 1.2 | 1.2 ± 1.5 | 7.5 ± 1.5* | 1.9 ± 1.5 | 0.0 ± 0.0* | 5.6 ± 1.2 | 14.4 ± 1.5 | 0.0 ± 0.0 |
| ICQ | 28.7 ± 4.6 | 1.9 ± 1.5 | 0.0 ± 0.0* | 3.8 ± 2.3 | 1.2 ± 1.5 | 1.2 ± 1.5 | 9.4 ± 2.0* | 2.5 ± 1.2 | 0.0 ± 0.0* | 42.5 ± 6.4 | 22.5 ± 3.1 | 0.6 ± 1.3* |
| OMAR | 28.7 ± 9.1 | 1.2 ± 1.5 | 0.0 ± 0.0* | 3.8 ± 1.2 | 0.6 ± 1.2 | 0.6 ± 1.2 | 0.6 ± 1.3 | 1.9 ± 1.5 | 0.0 ± 0.0* | 3.1 ± 0.0 | 11.9 ± 2.3 | 0.0 ± 0.0 |
| OMIGA | 40.6 ± 9.5 | 6.2 ± 5.6* | 0.0 ± 0.0* | 6.9 ± 1.2 | 2.5 ± 3.1 | 6.9 ± 1.2 | 5.6 ± 3.6* | 1.2 ± 1.5 | 0.0 ± 0.0* | 42.5 ± 6.4 | 22.5 ± 3.1 | 0.6 ± 1.3* |
| OptDICE | 37.5 ± 3.1 | 1.0 ± 1.5 | 0.0 ± 0.0* | 7.3 ± 3.9 | 0.0 ± 0.0 | 0.0 ± 0.0 | 0.0 ± 0.0 | 0.0 ± 0.0 | 0.0 ± 0.0* | 39.6 ± 5.3 | 11.9 ± 2.3 | 0.0 ± 0.0 |
| AlberDICE | 42.2 ± 6.4 | 1.6 ± 1.6 | 0.0 ± 0.0* | 3.9 ± 1.4 | 3.1 ± 0.0 | 0.0 ± 0.0 | 0.0 ± 2.6 | 2.3 ± 2.6* | 1.0 ± 1.5* | 43.1 ± 6.4 | 9.4 ± 6.8 | 0.0 ± 0.0 |
| ComaDICE | 55.0 ± 1.5 | 8.8 ± 7.0* | **0.6 ± 1.3** | 8.1 ± 3.2 | 7.5 ± 2.5 | 4.4 ± 4.2* | **11.2 ± 5.4** | 3.1 ± 2.0* | **1.9 ± 3.8** | 48.8 ± 2.5 | 27.3 ± 3.4 | 0.6 ± 1.3* |
| Oryx | **89.5 ± 3.6** | **9.0 ± 2.7** | 0.0 ± 0.0* | **27.3 ± 3.0** | **21.1 ± 1.4** | **6.8 ± 4.4*** | 9.7 ± 2.6* | **4.4 ± 1.7** | 0.0 ± 0.0* | **96.6 ± 3.0** | **52.7 ± 5.8** | **2.0 ± 2.6** |

Table 9: Results (Episode Return) on SMAC datasets from OG-MARL (Formanek et al., 2023a). Results from DoF (Li et al., 2025).

| | 3m | | | 8m | | | 5m_vs_6m | | | 2s3z | | | 3s5z_vs_3s6z | | |
|---|---|---|---|---|---|---|---|---|---|---|---|---|---|---|---|
| Algorithms | Good | Medium | Poor | Good | Medium | Poor | Good | Medium | Poor | Good | Medium | Poor | Good | Medium | Poor |
| MABCQ | 3.7 ± 1.1 | 4.0 ± 1.0 | 3.4 ± 1.0 | 4.8 ± 0.6 | 5.6 ± 0.6 | 3.6 ± 0.8 | 2.4 ± 0.4 | 3.8 ± 0.5 | 3.3 ± 0.5 | 7.7 ± 0.9 | 7.6 ± 0.7 | 6.6 ± 0.2 | 5.9 ± 0.3 | 6.5 ± 0.5 | 6.1 ± 0.6 |
| MACQL | 19.1 ± 0.1 | 13.7 ± 0.3 | 4.2 ± 0.1 | 5.4 ± 0.9 | 4.5 ± 1.5 | 3.5 ± 1.0 | 7.4 ± 0.6 | 8.1 ± 0.2 | 6.8 ± 0.1 | 17.4 ± 0.3 | 15.6 ± 0.4 | 8.4 ± 0.8 | 7.8 ± 0.5 | 8.5 ± 0.6 | 5.9 ± 0.4 |
| MAICQ | 18.7 ± 0.7 | 13.9 ± 0.8 | 8.4 ± 2.6 | **19.6 ± 0.2** | 17.9 ± 0.5 | 11.2 ± 1.3* | 11.0 ± 0.6 | 10.6 ± 0.6 | 6.6 ± 0.2 | 18.3 ± 0.2 | 17.0 ± 0.1* | 9.9 ± 0.6 | 13.5 ± 0.6 | 11.5 ± 0.2 | 7.9 ± 0.2* |
| MADT | 19.0 ± 0.3 | 15.8 ± 0.5 | 4.2 ± 0.1 | 18.5 ± 0.4 | 18.2 ± 0.1 | 4.8 ± 0.1 | 16.8 ± 0.1* | 16.1 ± 0.2* | 7.6 ± 0.3 | 18.1 ± 0.1 | 15.1 ± 0.2 | 8.9 ± 0.3 | 12.8 ± 0.2 | 11.6 ± 0.3 | 5.6 ± 0.3 |
| MADIFF | 19.3 ± 0.5* | 16.4 ± 2.6* | 10.3 ± 6.1* | 18.9 ± 1.1* | 16.8 ± 1.6 | 9.8 ± 0.9 | 16.5 ± 2.8* | 15.2 ± 2.6* | 8.9 ± 1.3 | 15.9 ± 1.2 | 15.6 ± 0.3 | 8.5 ± 1.3 | 7.1 ± 1.5 | 5.7 ± 0.6 | 4.7 ± 0.6 |
| DoF | 19.8 ± 0.2* | **18.6 ± 1.2** | 10.9 ± 1.1 | 19.6 ± 0.3* | 18.6 ± 0.8* | **12.0 ± 1.2** | **17.7 ± 1.1** | **16.2 ± 0.9** | 10.8 ± 0.3* | 18.5 ± 0.8 | **18.1 ± 0.9** | 10.0 ± 1.1 | 12.8 ± 0.8 | 11.9 ± 0.7 | 7.5 ± 0.2 |
| Oryx | **19.9 ± 0.2** | 17.4 ± 0.9* | **17.0 ± 1.2** | 19.3 ± 0.3 | **19.0 ± 0.3** | 5.1 ± 0.1 | 16.5 ± 0.7* | 15.5 ± 0.8* | **11.1 ± 0.4** | **19.7 ± 0.2** | 17.5 ± 0.5* | **12.0 ± 0.7** | **17.9 ± 0.5** | **13.0 ± 0.3** | **7.9 ± 0.1** |

Table 10: Results (Win Rate) on CFCQL (Shao et al., 2023) datasets. Other algorithm results also from CFCQL.

| Algorithm | 2s3z | | | | 3s_vs_5z | | | | 5m_vs_6m | | | | 6h_vs_8z | | | |
|---|---|---|---|---|---|---|---|---|---|---|---|---|---|---|---|---|
| | Medium | Med. Replay | Expert | Mixed | Medium | Med. Replay | Expert | Mixed | Medium | Med. Replay | Expert | Mixed | Medium | Med. Replay | Expert | Mixed |
| MACQL | 0.17 ± 0.08 | 0.12 ± 0.08 | 0.58 ± 0.34* | 0.67 ± 0.17 | 0.09 ± 0.06 | 0.01 ± 0.01 | 0.92 ± 0.05 | 0.17 ± 0.10 | 0.01 ± 0.01 | 0.16 ± 0.08* | 0.72 ± 0.05 | 0.01 ± 0.01 | 0.01 ± 0.01 | 0.08 ± 0.04 | 0.14 ± 0.06 | 0.01 ± 0.01 |
| MAICQ | 0.18 ± 0.02 | 0.41 ± 0.06 | 0.93 ± 0.04 | 0.85 ± 0.07 | 0.03 ± 0.01 | 0.01 ± 0.02 | 0.91 ± 0.04 | 0.10 ± 0.04 | 0.26 ± 0.03* | 0.18 ± 0.04* | 0.72 ± 0.03 | 0.67 ± 0.08 | 0.19 ± 0.04 | 0.04 ± 0.04 | 0.24 ± 0.08 | 0.05 ± 0.03 |
| OMAR | 0.15 ± 0.04 | 0.24 ± 0.09 | 0.95 ± 0.04* | 0.60 ± 0.04 | 0.00 ± 0.00 | 0.00 ± 0.00 | 0.64 ± 0.08 | 0.00 ± 0.00 | 0.19 ± 0.06 | 0.03 ± 0.02 | 0.33 ± 0.06 | 0.10 ± 0.10 | 0.04 ± 0.03 | 0.04 ± 0.03 | 0.12 ± 0.06 | 0.00 ± 0.00 |
| MADTKD | 0.18 ± 0.03 | 0.36 ± 0.07 | **0.99 ± 0.02** | 0.47 ± 0.08 | 0.01 ± 0.01 | 0.01 ± 0.01 | 0.67 ± 0.08 | 0.14 ± 0.08 | 0.21 ± 0.04 | 0.16 ± 0.04* | 0.58 ± 0.04 | 0.21 ± 0.05 | 0.22 ± 0.07 | 0.12 ± 0.05 | 0.48 ± 0.06 | 0.25 ± 0.07 |
| BC | 0.16 ± 0.07 | 0.33 ± 0.04 | 0.97 ± 0.02* | 0.44 ± 0.06 | 0.08 ± 0.02 | 0.01 ± 0.01 | 0.98 ± 0.02* | 0.21 ± 0.04 | 0.28 ± 0.37* | 0.18 ± 0.06* | 0.82 ± 0.04* | 0.21 ± 0.12 | 0.40 ± 0.03* | 0.11 ± 0.04 | 0.60 ± 0.04 | 0.27 ± 0.06 |
| IQL | 0.16 ± 0.04 | 0.33 ± 0.06 | 0.98 ± 0.03* | 0.19 ± 0.04 | 0.20 ± 0.05 | 0.04 ± 0.04 | **0.99 ± 0.01** | 0.20 ± 0.06 | 0.25 ± 0.02* | 0.18 ± 0.04* | 0.77 ± 0.03 | 0.76 ± 0.06* | 0.40 ± 0.05* | 0.17 ± 0.03* | 0.67 ± 0.03* | 0.36 ± 0.05 |
| AWAC | 0.19 ± 0.05 | 0.39 ± 0.05 | 0.97 ± 0.03* | 0.14 ± 0.04 | 0.19 ± 0.03 | 0.08 ± 0.05 | **0.99 ± 0.02** | 0.18 ± 0.03 | 0.22 ± 0.04 | 0.18 ± 0.04* | 0.75 ± 0.02 | **0.78 ± 0.02** | **0.43 ± 0.06** | 0.14 ± 0.04 | 0.67 ± 0.03* | 0.35 ± 0.06 |
| CFCQL | 0.40 ± 0.10* | **0.55 ± 0.07** | **0.99 ± 0.01** | 0.84 ± 0.09 | **0.28 ± 0.03** | 0.12 ± 0.04 | **0.99 ± 0.01** | 0.60 ± 0.14 | **0.29 ± 0.05** | **0.22 ± 0.06** | **0.84 ± 0.03** | 0.76 ± 0.07* | 0.41 ± 0.04* | **0.21 ± 0.05** | **0.70 ± 0.06** | 0.49 ± 0.08 |
| Oryx | **0.46 ± 0.08** | 0.17 ± 0.05 | **0.99 ± 0.02** | **0.98 ± 0.01** | 0.11 ± 0.05 | **0.33 ± 0.10** | 0.92 ± 0.03 | **0.79 ± 0.07** | 0.21 ± 0.05 | 0.21 ± 0.04* | 0.56 ± 0.06 | 0.50 ± 0.07 | 0.35 ± 0.10* | 0.13 ± 0.05 | 0.69 ± 0.07* | **0.62 ± 0.08** |

Table 11: Results (Episode Return) on RWARE datasets from Alberdice Matsunaga et al. (2023). Other algorithm results are also from AlberDICE.

| | Tiny (11x11) | | | Small (11x20) | | |
|---|---|---|---|---|---|---|
| Algorithm | N=2 | N=4 | N=6 | N=2 | N=4 | N=6 |
| BC | 8.80 ± 0.43 | 11.12 ± 0.33 | 14.06 ± 0.55 | 5.54 ± 0.10 | 7.88 ± 0.24 | 8.90 ± 0.23 |
| ICQ | 9.38 ± 1.30 | 12.13 ± 0.76 | 14.59 ± 0.28 | 5.43 ± 0.33 | 7.93 ± 0.33 | 8.87 ± 0.38 |
| OMAR | 6.77 ± 1.11 | 14.39 ± 1.58* | 16.13 ± 2.10* | 4.40 ± 0.59 | 7.12 ± 0.66 | 8.41 ± 0.85 |
| MADTKD | 6.24 ± 1.04 | 9.90 ± 0.36 | 13.06 ± 0.33 | 3.65 ± 0.59 | 6.85 ± 0.62 | 7.85 ± 0.90 |
| OptiDICE | 8.70 ± 0.10 | 11.13 ± 0.76 | 14.02 ± 0.62 | 4.84 ± 0.55 | 7.68 ± 0.16 | 8.47 ± 0.45 |
| AlberDICE | 11.15 ± 0.61 | 13.11 ± 0.55 | 15.72 ± 0.62 | 5.97 ± 0.19 | 8.18 ± 0.33 | 9.65 ± 0.23 |
| Oryx | **13.23 ± 0.25** | **16.71 ± 0.32** | **18.64 ± 0.47** | **6.95 ± 0.44** | **9.86 ± 0.32** | **10.13 ± 0.41** |

Table 12: Results (Normalised Episode Return) on MAMuJoCo datasets from OMAR (Pan et al., 2022). Other algorithm results from CFCQL (Shao et al., 2023).

| | 2x3 HalfCheetah | | | |
|---|---|---|---|---|
| Algorithm | Random | Medium-Replay | Medium | Expert |
| ICQ | 7.40 ± 0.00 | 35.60 ± 2.70 | 73.60 ± 5.00 | 110.60 ± 3.30 |
| TD3+BC | 7.40 ± 0.00 | 27.10 ± 5.50 | 75.50 ± 3.70 | 114.40 ± 3.80 |
| ICQL | 7.40 ± 0.00 | 41.20 ± 10.10 | 50.40 ± 10.80 | 64.20 ± 24.90 |
| OMAR | 13.50 ± 7.00 | 57.70 ± 5.10 | 80.40 ± 10.20* | 113.50 ± 4.30 |
| MACQL | 5.30 ± 0.50 | 37.00 ± 7.10 | 51.50 ± 26.70 | 50.10 ± 20.10 |
| IQL | 7.40 ± 0.00 | 58.80 ± 6.80 | 81.30 ± 3.70 | 115.60 ± 4.20 |
| AWAC | 7.30 ± 0.00 | 30.90 ± 1.60 | 71.20 ± 4.20 | 113.30 ± 4.10 |
| CFCQL | **39.70 ± 4.00** | 59.50 ± 8.20 | 80.50 ± 9.60 | 118.50 ± 4.90 |
| Oryx | 7.80 ± 7.80 | **91.70 ± 11.00** | **93.00 ± 10.90** | **128.70 ± 8.40** |

Table 13: Results (Episode Return) on MAMuJoCo datasets from OMIGA Wang et al. (2023). Other algorithm results from ComaDICE (Bui et al., 2025).

| | 3x1 Hopper | | | |
|---|---|---|---|---|
| **Algorithm** | Expert | Medium | Medium-Replay | Medium-Expert |
| BCQ | 77.90 ± 58.00 | 44.60 ± 20.60 | 26.50 ± 24.00 | 54.30 ± 23.70 |
| CQL | 159.10 ± 313.80 | 401.30 ± 199.90 | 31.40 ± 15.20 | 64.80 ± 123.30 |
| ICQ | 754.70 ± 806.30 | 501.80 ± 14.00 | 195.40 ± 103.60 | 355.40 ± 373.90 |
| OMIGA | 859.60 ± 709.50 | 1189.30 ± 544.30 | 774.20 ± 494.30* | 709.00 ± 595.70 |
| OptDICE | 655.90 ± 120.10 | 204.10 ± 41.90 | 257.80 ± 55.30 | 400.90 ± 132.50 |
| AlberDICE | 844.60 ± 556.50 | 216.90 ± 35.30 | 419.20 ± 243.50 | 515.10 ± 303.40 |
| ComaDICE | **2827.70 ± 62.90** | 822.60 ± 66.20 | 906.30 ± 242.10* | 1362.40 ± 522.90* |
| Oryx | 1811.70 ± 1269.70 | **2050.50 ± 927.10** | **1222.80 ± 544.60** | **1970.30 ± 1480.00** |

| | 2x4 Ant | | | |
|---|---|---|---|---|
| **Algorithm** | Expert | Medium | Medium-Replay | Medium-Expert |
| BCQ | 1317.70 ± 286.30 | 1059.60 ± 91.20 | 950.80 ± 48.80 | 1020.90 ± 242.70 |
| CQL | 1042.40 ± 2021.60* | 533.90 ± 1766.40* | 234.60 ± 1618.30* | 800.20 ± 1621.50* |
| ICQ | 2050.00 ± 11.90* | 1412.40 ± 10.90 | 1016.70 ± 53.50 | 1590.20 ± 85.60 |
| OMIGA | 2055.50 ± 1.60* | 1418.40 ± 5.40 | 1105.10 ± 88.90* | 1720.30 ± 110.60 |
| OptDICE | 1717.20 ± 27.00 | 1199.00 ± 26.80 | 869.40 ± 62.60 | 1293.20 ± 183.10 |
| AlberDICE | 1896.80 ± 33.70 | 1304.30 ± 2.60 | 1042.80 ± 80.80 | 1780.00 ± 23.60* |
| ComaDICE | 2056.90 ± 5.90* | 1425.00 ± 2.90* | 1122.90 ± 61.00* | 1813.90 ± 68.40* |
| Oryx | **2060.60 ± 8.90** | **1426.50 ± 7.00** | **1213.00 ± 113.20** | **1863.10 ± 118.90** |

| | 6x1 HalfCheetah | | | |
|---|---|---|---|---|
| **Algorithm** | Expert | Medium | Medium-Replay | Medium-Expert |
| BCQ | 2992.70 ± 629.70 | 2590.50 ± 1110.40* | -333.60 ± 152.10 | 3543.70 ± 780.90* |
| CQL | 1189.50 ± 1034.50 | 1011.30 ± 1016.90 | 1998.70 ± 693.90 | 1194.20 ± 1081.00 |
| ICQ | 2955.90 ± 459.20 | 2549.30 ± 96.30 | 1922.40 ± 612.90 | 2834.00 ± 420.30 |
| OMIGA | 3383.60 ± 552.70 | 3608.10 ± 237.40* | 2504.70 ± 83.50 | 2948.50 ± 518.90 |
| OptDICE | 2601.60 ± 461.90 | 305.30 ± 946.80 | -912.90 ± 1363.90 | -2485.80 ± 2338.40 |
| AlberDICE | 3356.40 ± 546.90 | 522.40 ± 315.50 | 440.00 ± 528.00 | 2288.20 ± 759.50 |
| ComaDICE | 4082.90 ± 45.70 | 2664.70 ± 54.20 | 2855.00 ± 242.20 | 3889.70 ± 81.60 |
| Oryx | **4784.20 ± 161.60** | **3753.40 ± 203.40** | **3521.10 ± 223.60** | **4182.70 ± 172.60** |

### E.1 Hyperparameters

**SMAC.** All Hyperparameter settings for Oryx were kept fixed across SMAC datasets except for the ICQ policy temperature, which we found was sensitive to the mean episode return of datasets. Thus, we used a policy temperature that ranged between 0.9 and 0.1, depending on the quality (mean episode return) of the respective datasets. For OG-MARL and OMIGA datasets, the policy temperature was set as follows: Good -> 0.9, Medium -> 0.3 and Poor -> 0.1. For CFCQL datasets, the policy temperature was set as follows: Medium -> 0.3, Med. Replay -> 0.1, Expert -> 0.9 and Mixed -> 0.8.

**RWARE.** All Hyperparameter settings for Oryx were kept fixed across RWARE datasets.

**MAMuJoCo.** All Hyperparameter settings for Oryx were kept fixed across MAMuJoCo datasets.

Table 14: Hyperparameters for Oryx on SMAC scenarios.

| **Parameter** | **Value** |
|---|---|
| Sample sequence length | 20 |
| Sample batch size | 64 |
| Learning rate | 3e-4 |
| ICQ Value temperature | 1000 |
| ICQ Policy temperature | $\in (0.1, 0.9)$ |
| Model embedding dimension | 64 |
| Number retention heads | 1 |
| Number retention blocks | 1 |
| Retention heads $\kappa$ scaling parameter | 0.9 |

Table 15: Hyperparameters for Oryx on RWARE scenarios.

| **Parameter** | **Value** |
|---|---|
| Sample sequence length | 20 |
| Sample batch size | 64 |
| Learning rate | 3e-4 |
| ICQ Value temperature | 10000 |
| ICQ Policy temperature | 0.1 |
| Model embedding dimension | 64 |
| Number retention heads | 1 |
| Number retention blocks | 1 |
| Retention heads $\kappa$ scaling parameter | 0.8 |

Table 16: Hyperparameters for Oryx on MAMuJoCo scenarios.

| **Parameter** | **Value** |
|---|---|
| Sample sequence length | 20 |
| Sample batch size | 64 |
| Learning rate | 3e-4 |
| ICQ Value temperature | 100000 |
| ICQ Policy temperature | 10 |
| Model embedding dimension | 64 |
| Number retention heads | 1 |
| Number retention blocks | 3 |
| Retention heads $\kappa$ scaling parameter | 0.9 |

