# OpenReview forum: "Oryx: a Scalable Sequence Model for Many-Agent Coordination in Offline MARL"
_NeurIPS.cc/2025/Conference — NeurIPS 2025 poster_

### Official Review · Reviewer_KGQ8 · 2025-06-28

**Clarity:** 3
**Significance:** 2
**Originality:** 2
**Rating:** 4
**Confidence:** 3

**Summary:**

The main methodological contribution of this work lies in integrating existing approaches—specifically, the combination of Sable (a retention-based sequence model) and MAICQ (implicit constraint Q-learning). The main claims made in the abstract and introduction accurately reflect the scope of this work. The paper clearly states that Oryx is a combination of existing methods (Sable and MAICQ) for scalable offline multi-agent reinforcement learning. While the work does not offer fundamentally new algorithmic insights, it provides a comprehensive and systematic empirical evaluation demonstrating the effectiveness and scalability of the combined approach across a wide range of challenging benchmarks.

**Questions:**

Is there any theoretical guarantee for the Oryx which is a combination of existing methods (Sable and MAICQ) for scalable offline multi-agent reinforcement learning?
The related work part can be further extended.

**Ethical Concerns:**

["NO or VERY MINOR ethics concerns only"]

**Final Justification:**

Thank you for the detailed rebuttal. I now better appreciate the non-trivial engineering contributions, particularly the novel autoregressive ICQ loss and the offline adaptation of Sable. The strong empirical results across diverse benchmarks, along with open-sourcing efforts, further strengthen the paper’s practical value.

While theoretical guarantees remain limited, I agree that the demonstrated performance and scalability justify sharing this work with the community. I have accordingly raised my score from 3 to 4. I encourage the authors to clarify the novelty and theoretical positioning more explicitly in future revisions.

**Limitations:**

The limitations are discussed explicitly in the conclusion. In addition to acknowledging that large-scale real-world validation remains future work, the paper implicitly recognizes that the core contribution is a thoughtful integration of existing techniques rather than a novel algorithmic breakthrough.

**Paper Formatting Concerns:**

no.

**Quality:**

3

**Strengths And Weaknesses:**

S. The discussion is candid about the scope and boundaries of the work.
W. The paper does not introduce fundamentally new algorithmic principles, but instead systematically demonstrates that this combination yields significant practical benefits in challenging offline MARL scenarios. The claims, experiments, and discussions throughout the paper accurately reflect this.

---

> ### Author Rebuttal · Authors · 2025-07-31
>
> We thank the reviewer for their time taken to review our paper. With the following response we hope to address the reviewers' concern regarding the novelty of our work as well as the question about theoretical scaling guarantees.
>
> **On the novelty and significance of Oryx**
>
> We understand that at a high level Oryx may seem like a simple combination of existing approaches; however, we would argue that this overlooks the many practical challenges we had to overcome in order to implement this algorithm successfully. Moreover, we believe that many significant and impactful contributions to science over the years have come from novel combinations of existing ideas (potentially from different subfields). Of course, if the results for Oryx were incremental at best, then one could justifiably lean towards rejecting it. But since we showed extensively that the combination of ideas in Oryx results in a substantial improvement over the state-of-the-art across a broad range of tasks, we believe that it's an important finding that should be shared with the research community. Particularly because scalable Offline MARL has huge practical potential for a range of problem areas.
>
> Furthermore, we think that overly optimising for novelty at scientific conferences has actually resulted in some bad incentives that have led to some dubious research. Take, for example, the recent Offline MARL benchmarking paper [1], which showed that many of the novel ideas introduced to the field in recent years as conference papers turned out not to perform as well as relatively simple algorithms that “simply” combined existing ideas from the literature. Thus, we think that penalising us for not appearing to be novel enough is unfair, particularly in this case where we have provided substantial empirical evidence that the method works far better than anything else in the field.
>
> **More details on Oryx contributions**
> Furthermore, would like to draw attention to some of the key aspects of our contribution, which we believe make it significant:
> * **Novel Sequential Autoregressive ICQ Loss**: We developed and formally derived a novel sequential autoregressive version of the ICQ loss, which we believe is a significant theoretical contribution.
> * **Offline Training of Online Algorithms**: A core challenge we addressed was modifying online (on-policy) algorithms like Sable and MAT to enable their effective offline training from off-policy samples from a replay buffer (dataset). This adaptation presented non-trivial technical hurdles.
> * **Network Architecture Modifications**: We carefully considered and implemented modifications to the original Sable (and MAT) network architecture to incorporate the dual Q-value and Policy distribution heads required for the ICQ loss.
> * **Temporal Memory Handling**: Appropriately handling the resetting of temporal memory in the retention-based memory of the Sable backbone, particularly when training from offline sequences, required careful attention to detail.
> * **Empirical Performance**: Our work demonstrates substantially superior performance, not just marginal, across multiple benchmarks (SMAC, RWARE, Multi-Agent MuJoCo) and new challenging datasets featuring up to 50 agents. Having addressed the issue raised by the reviewer regarding comparison to MAT, we are confident our robust empirical investigation differentiates our work within the existing literature.
> * **Open-Sourcing and Reproducibility**: To ensure full transparency and foster continued research, we have already made our implementation and the novel datasets openly available, complete with detailed instructions for experiment replication. We are confident this initiative will greatly facilitate further advancements stemming from Oryx-based models, and we eagerly anticipate the community's contributions building upon this foundation.
>
> **On Oryx’s theoretical guarantees**
> Sable, on which Oryx is built, has theoretical guarantees when using the PPO training objective. This stems from the work by Kuba et al. (2022) [2]. In particular, HAPPO (multi-agent PPO with autoregressive updates) was shown to be an instance of their multi-agent mirror learning framework and to be theoretically sound. To obtain an instance of mirror learning requires only defining a valid drift functional, neighbourhood operator and sampling distribution. In Sable, these design choices are exactly as they are for HAPPO, and therefore, Sable inherits the same theoretical monotonic improvement and convergence guarantees as HAPPO. However, although Oryx shares a network architecture that is similar in spirit as Sable, it is fundamentally different as an offline algorithm and uses a different autoregressive objective. This makes it non-trivial to place Oryx within the same mirror learning framework and thus claim similar theoretical guarantees. We consider this an interesting avenue for future work.
>
> While Oryx's theoretical guarantees remain an exciting avenue for future research, the significant empirical results and novel algorithmic contributions presented in this work already establish a strong foundation, demonstrating the considerable practical value and advancement this paper brings to the community.
>
> **In conclusion**, we trust we have addressed the main concerns raised by the reviewer. Should the reviewer have any follow-up questions, please do not hesitate to ask. We would be happy to engage during the author-reviewer discussion period.
>
> [1] Formanek, Claude, et al. “Dispelling the Mirage of Progress in Offline MARL through Standardised Baselines and Evaluation.” NeurIPS ‘24
>
> [2] Kuba, Jakub Grudzien, et al. "Heterogeneous-agent mirror learning: A continuum of solutions to cooperative marl." arXiv preprint arXiv:2208.01682 (2022).

---

### Official Review · Reviewer_wEyu · 2025-06-28

**Clarity:** 2
**Significance:** 2
**Originality:** 2
**Rating:** 3
**Confidence:** 5

**Summary:**

This work proposes Oryx, which combines (1) Sable for sequential modeling, (2) ICQ for mitigating value over-estimation and (3) COMA in Offline MARL. Oryx is compared with other baselines on MA-MuJoCo, SMACv1 and RWARE as well as Connector and T-Maze.

**Questions:**

Please address each point raised in Strengths and Weaknesses.

**Ethical Concerns:**

["NO or VERY MINOR ethics concerns only"]

**Final Justification:**

The authors have added additional experimental results which partially resolved my concerns regarding its comparison to MAT which is another CTCE algorithm.
However, I believe the decision to mix CTCE and CTDE in the experimental setup makes the claims in the paper much weaker and inconclusive.
Simply put, the policies used in Oryx have more information (joint observations + joint actions) compared to baselines (local observations + local actions). It is not an apples to apples comparison.

Furthermore, as the other reviewers have also pointed out, the algorithmic contributions are minimal.
As a combination of advantage decomposition and ICQ, the theoretical contributions are not significant. The current form of the paper is lacking some key insight which explains why this combination is non-trivial (algorithmically) and why it is better than other methods, especially CTCE methods.

I recommend making this distinction clear and using a fair setup if the authors still want to emphasize algorithmic contributions. Otherwise, the dataset itself can be a contribution in a more appropriate venue.

My overall impression of the paper has improved, but I don't find it ready for publication in its current form.
Thus, I've raised my score on originality and significance from 1 to 2 and the final score from 2 to 3.

**Limitations:**

yes.

**Paper Formatting Concerns:**

None.

**Quality:**

2

**Strengths And Weaknesses:**

The major weakness is in the use of autoregressive policies, while baselines use independent policies in the experiments. This choice is not about the different algorithmic choices (e.g. sequential updates as in HAPPO), but about the policy structure itself. This policy structure should be the same for all algorithms for a fair comparison. I believe the authors are confusing autoregressive policy structures $\pi(a|s) = \pi_1(a_1|s) \pi_2(a_2|s, a_1) \cdots$ on the one hand and sequential policy updates where each agent learns a best response to the other agents’ current policies. If Oryx is the only algorithm which uses an autoregressive policy structure, then its an unfair comparison with other baselines which use an independent policy structure i.e. $\pi(a|s) = \pi_1(a_1|s) \pi_2(a_2|s) \cdots$.

Second, as far as I can decipher from the architecture in Figure 1, Oryx requires joint observations in the encoder during both training and testing, which means it is not Centralized Training Decentralized Execution (CTDE) but rather Centralized Training Centralized Execution (CTCE). It is a little strange that Oryx was compared to different CTDE algorithms while not compared with MAT [1] which is CTCE.

Finally, Oryx is simply a combination of three pre-existing algorithms, namely Sable, ICQ and COMA.

In terms of strengths, the set of benchmarks and baselines considered is comprehensive. However, the experimental setup is not correct as aforementioned, with a lack of clarity between CTCE vs CTDE and autoregressive policy structures.

[1] Multi-Agent Reinforcement Learning is a Sequence Modeling Problem (Wen, et al. NeurIPS 2022)

---

> ### Author Rebuttal · Authors · 2025-07-31
>
> We thank the reviewer for the time taken to review our paper. We appreciate the reviewer highlighting that our “set of benchmarks and baselines considered is comprehensive.”
>
> **Comparing Oryx to MAT**
>
> We acknowledge that it may not be immediately clear that our algorithm is not a CTDE algorithm and thank the reviewer for pointing this out; by no means did we intend to obscure this fact. We thought it was clear that by building on the MAT and Sable line of work, the reader would be aware that Oryx is a centralised approach. In hindsight, this assumption was misguided.
>
> Our first iteration of the Oryx algorithm was built by extending the MAT network architecture (rather than the Sable one), but ultimately, we settled on the variant with the Sable network architecture because of its superior performance and scaling properties. But since Sable had already been shown to be better than MAT (at least in the online case), we thought it was not worth repeating here. We therefore only reported our performance compared to algorithms from the existing offline literature, and to date, all but MADIFF are CTDE algorithms.
>
> Having now considered your review, we see that this was a mistake and can easily lead to confusion. To address this, we will add in our experiments comparing Oryx to the variant which used the MAT network architecture, and be clear about their difference to CTDE algorithms. Below are the results across all 43 SMAC datasets and all 6 RWARE datasets. On each dataset, we ran each algorithm over 10 random seeds.
>
> However, converting MAT, an online and on-policy algorithm, to an offline algorithm was itself a non-trivial task, which involved
> * Replacing the learning from a queue of online samples with learning from sampled batches of experience from a replay buffer (dataset).
> * Replacing the PPO loss  (which only works with online samples) with a loss function that works on off-policy samples.
> * Modifying the MAT network architecture to accommodate the new loss.
>
> In particular, we replaced the PPO-style loss in MAT with our novel Autoregressive version of the ICQ loss that we derived (and also use in Oryx). The MAT network modifications required for this were similar to the ones outlined in the Oryx paper.
>
> **Aggregated Results: Oryx vs. MAT**
>
> To compare the MAT variant and Oryx in a robust way, we applied the proposed statistical methodology from MARL-eval [1]. Namely, we provide 4 different aggregated metrics: the mean, median, interquartile mean, and optimality gap, with stratified bootstrap confidence intervals [in square braces]. To facilitate aggregation, the results for each dataset were normalised against the highest performance on that dataset before computing the aggregated performance metrics. From the tables below, it is clear to see that Oryx substantially outperforms the offline MAT+ICQ on both SMAC and RWARE datasets, with better scores across all metrics (for optimality gap, a lower score is better). We will add these results, along with a clarification of the difference between these methods and the CTDE methods, to our revised paper.
>
> **SMAC**
> |                             | MAT+ICQ            | ORYX                   |
> |------------------------|--------------------------|-------------------------|
> | Median                | 0.71 [0.62, 0.74]    | 0.91 [0.88, 0.92]   |
> | IQM                     | 0.67 [0.66, 0.69]    | 0.87 [0.86, 0.88]   |
> | Mean                   | 0.63 [0.62, 0.64]    | 0.77 [0.76, 0.79]   |
> | Optimality Gap    |  0.38 [0.36, 0.39]    | 0.23 [0.22, 0.25]   |
>
> **RWARE**
> |                             | MAT+ICQ            | ORYX                 |
> |------------------------|--------------------------|------------------------|
> | Median                | 0.85 [0.84, 0.86]    | 0.89 [0.88, 0.9]    |
> | IQM                     | 0.85 [0.84, 0.86]    | 0.89 [0.89, 0.9]    |
> | Mean                   | 0.84 [0.83, 0.84]    | 0.89 [0.88, 0.9]    |
> | Optimality Gap    | 0.16 [0.16, 0.17]    | 0.11 [0.1, 0.12]    |
>
> **Raw Tabulated Results**
>
> We will include the raw tabulated results for these two baselines alongside the others in the appendix of the camera-ready version of the paper. We could not include them here because the tables used too many characters, given the character limit. From the results, one can see that MAT+ICQ is also a very strong algorithm when compared to the CTDE algorithms from the literature. Which, we agree, is to be expected given their Autoregressive action selection at evaluation time. To us, this result highlights the benefit of using this class of algorithms whenever possible, i.e., when fully decentralised execution is not required (more on this towards the end of our rebuttal). Our work is also the first to demonstrate the applicability of autoregressive action selection in the offline MARL setting.
>
>
> **On the Significance of our Contribution**
>
> We believe that developing Oryx involved several intricate steps and novel insights that extend beyond a simple integration of existing ideas. We would like to highlight the following key aspects:
> * **Novel Sequential Autoregressive ICQ Loss**: We developed and formally derived a novel sequential autoregressive version of the ICQ loss, which we believe is a significant theoretical contribution.
> * **Offline Training of Online Algorithms**: A core challenge we addressed was modifying online (on-policy) algorithms like Sable and MAT to enable their effective offline training from off-policy samples from a replay buffer (dataset). This adaptation presented non-trivial technical hurdles.
> * **Network Architecture Modifications**: We carefully considered and implemented modifications to the original Sable (and MAT) network architecture to incorporate the dual Q-value and Policy distribution heads required for the ICQ loss.
> * **Temporal Memory Handling**: Appropriately handling the resetting of temporal memory in the retention-based memory of the Sable backbone, particularly when training from offline sequences, required careful attention to detail.
> * **Empirical Performance**: Our work demonstrates substantially superior performance, not just marginal, across multiple benchmarks (SMAC, RWARE, Multi-Agent MuJoCo) and new challenging datasets featuring up to 50 agents. Having addressed the issue raised by the reviewer regarding comparison to MAT, we are confident our robust empirical investigation differentiates our work within the existing literature.
> * **Open-Sourcing and Reproducibility**: To further contribute to the field and facilitate future research, we have already open-sourced both our implementation and the novel datasets, along with clear instructions for reproducing our experiments. We are confident that this will pave the way for further innovation in models derived from Oryx, and we are excited about the potential for others to build upon this direction.
>
> Moreover, we believe that classifying our contribution as merely "combining existing ideas" might overlook a common aspect of scientific advancement. Many significant breakthroughs often emerge from novel syntheses of existing concepts, and Oryx exemplifies this by integrating diverse ideas from the MARL field to achieve a result that is greater than the sum of its individual components. While an incremental improvement on a narrow set of benchmarks might warrant such a conclusion, our work demonstrates a significant performance gain over existing methods, supported by extensive experimentation. Furthermore, we have expanded the scope of research by developing and sharing a challenging set of benchmark datasets with up to 50 agents. Having addressed the experimental oversight highlighted by the reviewer, we are confident that our method represents a novel and significant contribution.
>
> **Final Note on CTDE vs Oryx (and MAT/Sable)**
>
> Indeed, our algorithm is not a CTDE algorithm; it belongs to the same class of algorithms as MAT and Sable. Thus, it is not directly applicable in settings where decentralised execution is required (although it's always possible to apply a decentralised policy distillation step at the end of training, if required [2,3]). Having said that, algorithms like MAT, Sable, and Oryx are nonetheless still practically very valuable as they lie somewhere in between the two extremes of fully independent actors (which scales well but may converge on a sub-optimal joint policy) and fully centralised learners (which scale poorly but have better guarantees to converge on an optimal policy in the limit) [4]. Thus, in problems with a large joint action space, which can naturally be factorised, Oryx (as well as MAT and Sable) provides a solution that trades off scalability on the one hand and quality of the joint policy on the other. For this reason, Oryx remains a highly relevant algorithm for many practical industrial applications.
>
> **In conclusion**, we once again thank the reviewer for their valuable input on our paper. Having addressed the main concern we feel the work is significantly strengthened. Should the reviewer have any follow-up questions, please do not hesitate to ask. We would be happy to engage during the author-reviewer discussion period.
>
> [1] Gorsane, Rihab, et al. “Towards a Standardised Performance Evaluation Protocol for Cooperative MARL.” NeurIPS ‘22
>
> [2] Zhu, Zhengbang, et al. “MADiff: Offline Multi-agent Learning with Diffusion Models,”
> NeurIPS ‘24
>
> [3] Li, Yueheng, et al. “Multi-Agent Guided Policy Optimization.” Arxiv 2025
>
> [4] de Kock, Ruan, et al. “Is an Exponentially Growing Action Space Really That Bad? Validating a Core Assumption for using Multi-Agent RL.” AAMAS ‘25

---

> > ### Comment · Reviewer_wEyu · 2025-08-04
> >
> > Thank you for the rebuttal.
> >
> > I still think the algorithmic contributions are minimal, and it is better to re-consider the positioning of the paper, perhaps for the Datasets and Benchmarks Track, focusing on the open-source release of datasets. While I appreciate the  additional experiments with MAT, it also showed that it is on-par with Oryx on RWARE which is the more realistic environment requiring complex coordination. To me this shows that (1) there is a fundamental limitation of using ICQ due to the IGM assumption used for value decomposition and (2) the performance gain in Oryx came primarily from the CTCE nature of its algorithm.
> >
> > Thus, I will maintain my score.

---

> ### Author Response · Authors · 2025-08-04
> **Possible misinterpretation of RWARE results**
>
> Dear reviewer,
>
> We believe that you have misinterpreted the RWARE results. Our results show that Oryx outperforms MAT+ICQ across all 4 metrics with no overlap in confidence interval [1]. Therefore, we can conclude with statistical significance that Oryx is superior to MAT+ICQ on RWARE.
>
> With the additional experimental evidence showing that Oryx is superior to MAT+ICQ, which are both offline CTCE methods, we have addressed your only stated concern with the methodology of our paper. Therefore, we feel strongly that your rating of 2 is no longer justified. For instance, a rating of 2 implies that our paper has “technical flaws, weak evaluation, inadequate reproducibility, and incompletely addressed ethical considerations.”
>
> * **Technical flaws:** None have been raised.
> * **Weak evaluation:** We have resolved the only weakness mentioned by your review. Furthermore, the breadth of benchmarks and baselines is far greater than the widely accepted standard in the field [2]. We tested on over 88 datasets, from 5 different environments. We covered all the recent SOTA offline algorithms as well as demonstrated superior performance to a robust implementation of a novel baseline MAT+ICQ.
> * **Inadequate reproducibility:** We publicly release all of our code and datasets with clear instructions on how to reproduce our results. This is once again a substantial improvement over the widely accepted state of the field [2].
> * **Incompletely addressed ethical considerations:** clearly does not apply.
>
> Finally, with regards to your statement that “(1) there is a fundamental limitation of using ICQ due to the IGM assumption used for value decomposition”, we are not entirely sure what you mean, but if you are referring to the QMIX component in the original MAICQ algorithm, then we will answer as follows. The variant of the ICQ loss that we derive in this paper builds on the single-agent version of the ICQ loss that is introduced in the first half of the paper [3], not the MAICQ loss, which includes QMIX (value decomposition). Please make sure to see our proof in the appendix of the paper for all the details on the derivation. **Thus, our autoregressive version of the ICQ loss should be seen as another way to extend the original ICQ loss to the multi-agent setting, entirely separate to the value decomposition variant (MAICQ), which relies on the IGM principle.**
>
> If, however, we misunderstood what you are trying to say, please do elaborate so that we can accurately provide a response.
>
> [1] Agarwal, Rishabh, et al. “Deep Reinforcement Learning at the Edge of the Statistical Precipice”, NeruIPS ‘21
>
> [2] Claude Formanek, et al. “Dispelling the Mirage of Progress in Offline MARL through Standardised Baselines and Evaluation”, NeurIPS ‘24
>
> [3] Yang, Yiqin, et al. “Believe What You See: Implicit Constraint Approach for Offline Multi-Agent Reinforcement Learning”, NeurIPS ‘21

---

> > ### Comment · Reviewer_wEyu · 2025-08-04
> >
> > Thanks for the clarifications.
> > I understand the ICQ loss part, which was a misunderstanding on my part. It seems that there is no IGM assumption as in MAICQ. Even so, I believe that it is a combination of advantage decomposition and ICQ, and the theoretical contributions are not significant. This is fine but it is lacking some key insight which explains why this combination is non-trivial (algorithmically) and why it is better than other methods, especially CTCE methods.
> >
> > For the extra experimental results, what are the tasks and datasets used for RWARE and SMAC?
> > Even if the datasets and tasks covered in the paper is comprehensive, my concern was about CTCE vs CTDE. Positive results for one task and one dataset vs one baseline do not warrant a claim that Oryx is definitively better than other CTCE approaches. There are plenty of other baselines which can be considered as well such as an autoregressive policy version of the CTDE baselines.

---

> > > ### Author Response · Authors · 2025-08-04
> > > **Misunderstanding about the tasks and datasets we used in our experiments**
> > >
> > > Dear Reviewer,
> > >
> > > I think there may be another misunderstanding. We did not conduct the additional experiments on a “single task and dataset”. **We conducted the experiments over all the tasks and datasets in RWARE and SMAC (15 tasks and 49 datasets).** As we outlined in the paper, for RWARE that includes the tasks `tiny-2ag`, `tiny-4ag`, `tiny-6ag`, `small-2ag`, `small-4ag` and `small-6ag`. For SMAC, it includes the tasks `3m, 8m, 5m_vs_6m, 2s3z, 3s5z_vs_3s6z, 6h_vs_8z, 2c_vs_64zg, 3s_vs_5z` and `corridor`. The two tables we provided in our rebuttal are the aggregated metrics across all of these tasks. This ensures that we can make statistically sound conclusions about the performance of Oryx in these environments. We refer the reviewer to the following two papers for more details on the value of aggregate results [1,2]. Because we are drawing our conclusion that Oryx is better than MAT+ICQ from such a broad set of results, we have high confidence that we are correct.
> > >
> > > With regard to the reviewer's request for an additional baseline that is “an autoregressive policy version of the CTDE baselines”, we believe that constructing such a baseline is not possible since an autoregressive policy ($\pi(a|s)=\pi(a_1|s)\pi(a_2|s,a_1)...)$ as the reviewer put it) implies the algorithm does not permit decentralised execution. But if we have misunderstood what the reviewer is asking for, please don’t hesitate to expand so that we can try to address your concern.
> > >
> > > [1] Agarwal, Rishabh, et al. “Deep Reinforcement Learning at the Edge of the Statistical Precipice”, NeruIPS ‘21
> > >
> > > [2] Gorsane, Rihab, et al. “Towards a Standardised Performance Evaluation Protocol for Cooperative MARL.” NeurIPS ‘22

---

> > > > ### Comment · Reviewer_wEyu · 2025-08-05
> > > >
> > > > Thanks for the clarifications on the experiments.
> > > >
> > > > In terms of autoregressive policies, this is a matter of choice of policy class used for each algorithm. In principle, I believe every baseline used for Figure 4 (MAICQ, ICQ, IQL, AWAC, BC, ...) can be re-trained with an autoregressive policy. In essence this will make the experimental setup more fair since Oryx is using CTCE.

---

> > > > > ### Author Response · Authors · 2025-08-05
> > > > > **On adding autoregressive policies to other baselines from the literature**
> > > > >
> > > > > Dear reviewer,
> > > > >
> > > > > Thank you for the clarification. If we understand correctly, you are asking that we take an existing CTDE offline MARL method from the literature and modify it such that its policy is autoregressive. This is quite different from the reviewers' original request from the review, which was to include an offline version of MAT as a baseline (which we have now provided in our rebuttal).
> > > > >
> > > > > To modify standard CTDE algorithms to be autoregressive would require modifying the policy networks to be able to handle dynamically changing inputs based on how many prior agents' actions need to be considered. Since most of these methods use standard MLPs, this could technically be achieved with task-specific masking. However, this would be a hack, whereas sequence models like MAT and Sable are already well-suited to handle these kinds of inputs and represent a natural model class for such policies.  Designing an autoregressive policy network, from within a model class that is not naturally suited for such a design, did not make sense to us (and still doesn’t), and we have a strong prior belief that it won’t work well.
> > > > >
> > > > > We think that extending the sequence modelling line of work from MAT and Sable, to the Offline MARL setting is clearly a very valuable thing to do. Moreover, we are confident that we have done it in a compelling and robust way. By deriving an autoregressive version of the ICQ loss, we developed a principled way of training autoregressive multi-agent sequence models, like our modified versions of the MAT and Sable networks, on offline data. We showed conclusively that our MAT+ICQ model and our flagship Oryx model outperform the existing CTDE methods in the literature. Validating the utility of these methods when applicable to one's problem (i.e. decentralised execution is not required).

---

> > > > > > ### Comment · Reviewer_wEyu · 2025-08-06
> > > > > >
> > > > > > > If Oryx is the only algorithm which uses an autoregressive policy structure, then its an unfair comparison with other baselines which use an independent policy structure
> > > > > >
> > > > > > This is quoted from my original review, so this concern is not something new.
> > > > > >
> > > > > > For instance, take BC which is the most simple baseline.
> > > > > > This can easily be extended to autoregressive policies of the form $\pi_1(a_1|s) \pi_2(a_2|s, a_1) $  by changing the input of the policy to also contain the actions by previous agent actions. If using shared parameters, assuming each agent has the same action dimensions, the additional input to the policy can be size $|A_i| \times (N-1)$, where the inputs are masked out accordingly. Similarly, just simply using $|A_i| \times (i-1)$ size inputs if using separate parameters.  During training, the batch contains joint actions, so we simply use the sampled previous agent actions as input. During execution, the agents will act sequentially conditioning on the actions by previous agents, as you do in Oryx.
> > > > > >
> > > > > > As analyzed in [1], the autoregressive policy structure can represent any joint policy, so it is naturally much more expressive than independent policies. By using autoregressive policies, it is diverging from the traditional notion of CTDE. While consideration of CTCE is fine, the authors' claim that Oryx outperforms all baselines is inconclusive, since we can convert all baselines to handle autoregressive policies.
> > > > > >
> > > > > > [1] Revisiting Some Common Practices in Cooperative Multi-Agent Reinforcement Learning (Fu, et al. ICML 2022)

---

> > > > > > > ### Author Response · Authors · 2025-08-06
> > > > > > > **Lack of strong evidence for the proposed AR policy approach**
> > > > > > >
> > > > > > > Dear reviewer,
> > > > > > >
> > > > > > > We thank you for highlighting the work in [1]. We find this interesting and will include it in our paper when discussing the architectural and algorithmic design around auto-regressive policies.
> > > > > > >
> > > > > > > However, if you carefully consider the results in the paper, you will notice that only three higher-dimensional tasks were considered for autoregressive policies, namely, 2m_vs_1z and 3s_vs_5z from SMAC, as well as 3v1 in GRF (Table 9). Importantly, these tasks have very few agents, i.e. 2, 3, and 3 agents respectively. Moreover, while independent policies were not tested on 2m_vs_1z, it was on 3s_vs_5z, and its performance was the same as the AR policy (Table 4). On 3v1 GRF,  the mean performance of AR policies is slightly better than independent policies but with largely overlapping standard deviations, and is therefore unlikely to be a statistically significant result (Table 5). Finally, the authors themselves say: “In general, when optimizing the final reward is not the only goal of a research project, we would suggest adopting auto-regressive modeling for diverse emergent behaviors.” Thus, the authors themselves seem to suggest that these AR models tend to underperform independent policy methods and should be considered for use cases other than maximising reward.
> > > > > > >
> > > > > > > Therefore, the referenced paper does not provide enough evidence that implementing AR policies (in the way suggested) will work well for arbitrary algorithms. More specifically, the paper lacks evidence for larger numbers of agents (n>3). Whereas a core contribution of our work focuses on strong performance at scale (up to 50 agents).
> > > > > > >
> > > > > > > Therefore, in conclusion, we thank the reviewer for their insight on AR policies, however, we maintain our view that converting all of the existing baselines in offline MARL to use this type of policy will add little value to our work and will not change our core findings. Adding MAT was indeed a key missing baseline within our study, and we thank the reviewer for pointing this out, but with it, we believe your two main concerns regarding an unfair comparison of (1) CTDE vs CTCE and (2) AR vs not, should be adequately addressed. In (1), we already include another CTCE baseline from prior accepted work, MADIFF [2], and for (2), we include a strong AR baseline, MAT, that has a natural architectural alignment with such a policy class.
> > > > > > >
> > > > > > > [1] Revisiting Some Common Practices in Cooperative Multi-Agent Reinforcement Learning (Fu, et al. ICML 2022)
> > > > > > >
> > > > > > >
> > > > > > > [2] Zhu, Zhengbang, et al. “MADiff: Offline Multi-agent Learning with Diffusion Models,” NeurIPS ‘24

---

### Official Review · Reviewer_vYPS · 2025-07-02

**Clarity:** 3
**Significance:** 3
**Originality:** 3
**Rating:** 4
**Confidence:** 5

**Summary:**

Oryx proposes a novel offline multi-agent reinforcement learning (MARL) algorithm addressing two core challenges: extrapolation error and miscoordination in many-agent settings. It integrates:
1. Sable's retention-based architecture for long-context sequence modeling.
2. Sequential Implicit Constraint Q-learning (ICQ) with auto-regressive policy updates.
3. Counterfactual advantage for low-variance policy gradients.

Evaluated across various datasets (SMAC, RWARE, Multi-Agent MuJoCo), Oryx achieves SOTA in >80% of tasks, demonstrating superior scalability and coordination.

**Questions:**

1. How does inference memory scale with agent count (e.g., in 50-agents)?

2. Can sequential updates handle highly heterogeneous agents?

**Ethical Concerns:**

["NO or VERY MINOR ethics concerns only"]

**Final Justification:**

My questions have been solved.

**Quality:**

3

**Strengths And Weaknesses:**

Strengths:

1. Novel fusion of Sable (retention mechanism) and sequential ICQ enables stable many-agent coordination. Sequential policy updates conditioned on prior agents' actions explicitly mitigate miscoordination.

2. Extensive benchmarks spanning discrete/continuous control, agent scales, and horizon lengths. Ablations validate core components (e.g., T-Maze proves necessity of auto-regression, memory, and ICQ).

3. Dominates in dense-agent settings (e.g., 50-agent Connector), where prior methods (MAICQ) fail. Introduces new large-scale datasets for community use.

Weaknesses

1. Retention blocks and sequential decoding likely increase training costs vs. non-sequential methods (not quantified).

---

> ### Author Rebuttal · Authors · 2025-07-31
>
> We thank the reviewer for their constructive comments on our paper. We appreciate the reviewer highlighting the novelty of our algorithm and acknowledging that the new benchmark datasets we provide are valuable contributions. Concerning the author's questions, we now address each in detail:
>
> * *How does inference memory scale with agent count (e.g., in 50-agents)?*
>     - The Sable [1] paper showed the scaling properties of the network for agents up to 1000. The memory scaling properties are significantly better than MAT (which is quadratic in the number of agents) and competitive with algorithms that use independent actors. This suggests that Oryx will scale to even more than 50 agents. The main bottleneck we encountered with scaling beyond 50 agents was handling the very large datasets. Generating datasets for systems with so many agents became quite an engineering challenge to handle. We stopped at 50 for this paper because we felt it struck a good balance between pushing the frontier of research while still being manageable for other future researchers to use without too much engineering overhead. In the future, building a better data pipeline should allow us to test Oryx at even greater scale.
> * *Retention blocks and sequential decoding likely increase training costs vs. non-sequential methods (not quantified).*
>     - Yes, it is true that sequential decoding increases computational complexity in the number of agents. In MAT, the complexity scales quadratically in the number of agents, which can be prohibitive for large numbers of agents. Sable, on the other hand, has far superior scaling properties thanks to the retentive network architecture. While not as good as non-sequential (independent) action selection methods, it is very competitive up to 1000 agents. Sable (and by extension Oryx) therefore strikes a good balance between superior performance and competitive scaling. It is for this reason that we chose to use the Sable network as a starting point for Oryx rather than the MAT network.
> * *Can sequential updates handle highly heterogeneous agents?*
>     - Yes, it can! Although how to achieve this depends on what exactly you mean by heterogeneous agents. If you mean the agents assume different roles in the environment, but their observation and action spaces are the same, then Oryx handles this gracefully right out of the box. If, on the other hand, agents have different-sized observation spaces and action spaces, you can still apply Oryx, but first need to pad the spaces to the same size. This is relatively easy to do and has worked well when we have tried it. Moreover, we are confident about Oryx’s ability to handle heterogeneous agents because it builds on the foundations set by the Heterogeneous Agents RL [3] line of work, which first introduced the sequential updating scheme that was then extended to Sequence Modelling by MAT [2], and finally Sable.
>
> Once again, we thank the reviewer for their comments. Should the reviewer have any follow-up questions, please do not hesitate to ask. We would be happy to engage during the author-reviewer discussion period.
>
> [1] Mahjoub, Omayma, et al. “Sable: a Performant, Efficient and Scalable Sequence Model for MARL.” ICML ‘25
>
> [2] Wen, Muning, et al. “Multi-Agent Reinforcement Learning is a Sequence Modeling Problem.” NeurIPS ‘22
>
> [3] Zhong,Yifan, et al. “Heterogeneous-agent reinforcement learning.” Journal of Machine Learning Research, January ‘24

---

> > ### Comment · Reviewer_vYPS · 2025-08-06
> >
> > I appreciate the reponse and rebuttal, which has cleared my concerns.

---

### Official Review · Reviewer_MxbN · 2025-07-06

**Clarity:** 4
**Significance:** 4
**Originality:** 3
**Rating:** 5
**Confidence:** 3

**Summary:**

This paper introduces Oryx, a novel algorithm designed for offline multi-agent reinforcement learning (MARL), targeting the key challenges of extrapolation error and miscoordination that arise when learning from fixed datasets in multi-agent environments. Oryx builds on Implicit Constraint Q-learning (ICQ) and incorporates the Advantage Decomposition Theorem to design a sequential policy updating scheme, where agents update their policies one at a time, holding others fixed. The method also employs a modified version of the Sable network architecture to better handle the joint action-value function and improve credit assignments.

Empirical evaluations are conducted on diverse and challenging benchmarks, including SMAC, RWARE, Multi-Agent MuJoCo. The results demonstrate that Oryx achieves state-of-the-art performance, particularly in settings with high agent density and complex coordination demands. The authors also released new large-scale offline MARL datasets to support future research.

**Questions:**

1. While sequential updates may help coordination, do they scale well in environments with hundreds of agents or continuous control tasks? Could you discuss whether this design limits parallelization or increases training time significantly? I believe it is worth mentioning and discussing the trade-offs and limitations.

2. In the sequential update scheme, how sensitive is the overall policy quality to the order in which agents are updated? Have you tried optimizing the order, and does it affect stability or final performance?

3. In sequential updates, each agent optimizes assuming the policies of the other agents are fixed. Does this introduce error accumulation across the agent chain? Is there any correction mechanism or re-evaluation strategy to mitigate this?

4. In your implementation of Implicit Constraint Q-learning for each agent, how is the constraint enforced during policy updates? Are constraints handled through soft penalties or something else? Additional clarity would help.

**Ethical Concerns:**

["NO or VERY MINOR ethics concerns only"]

**Final Justification:**

The authors have provided thoughtful responses to my concerns. They clarified the scalability of the sequential update scheme and explained how constraints are handled in ICQ. They also pointed out their T-Maze ablation study, which isolates the role of each component. While the primary contribution is a well-engineered integration of known ideas, the empirical performance and release of datasets make it a good addition to the offline MARL literature.

**Limitations:**

No. While the authors mention some future work, they do not explicitly discuss the limitations or trade-offs of the proposed method. Specifically, the paper would benefit from a discussion of the scalability bottlenecks or other trade-offs introduced by sequential policy updates.

**Quality:**

3

**Strengths And Weaknesses:**

## Strengths:

 + The proposed method combines Implicit Constraint Q-learning and the Advantage Decomposition Theorem to address critical challenges in offline MARL, such as extrapolation error and agent miscoordination. The use of sequential policy updates is well-motivated.

 + Oryx demonstrates state-of-the-art results across multiple offline MARL benchmarks, including high-agent-density tasks like Connector and SMAC, showcasing the method’s scalability and effectiveness.

 + The release of open-source implementations and offline datasets in complex multi-agent environments provides a valuable resource for the community.

## Weaknesses:

 - The paper lacks ablation studies to isolate the contribution of each component (e.g., ICQ, Sable architecture, sequential updates). It is unclear which aspects of the method drive performance gains.

 - While the sequential update strategy improves coordination, it may become computationally expensive or impractical in very large-scale or real-time settings. The paper does not fully address the trade-offs or potential bottlenecks.

---

> ### Author Rebuttal · Authors · 2025-07-31
>
> We thank the reviewer for their positive review of our work and their constructive questions. By addressing them, we feel we can substantially improve the clarity of our paper's message. Here we address each of your questions in turn:
>
> * *While sequential updates may help coordination, do they scale well in environments with hundreds of agents or continuous control tasks?*
>     - Similar to our response to the reviewer vYPS, since Oryx uses a modified version of the Sable network architecture, it inherits many of its scaling properties. In particular, the Sable paper showed superior scaling trends to MAT [1] and competitive trends compared to fully independent actors for up to 1000 agents. This suggests that Oryx will scale to even more than 50 agents. The main bottleneck we encountered with scaling beyond 50 agents was handling the very large datasets. Generating datasets for systems with so many agents became quite an engineering challenge to handle. We stopped at 50 for this paper because we felt it struck a good balance between pushing the frontier of research while still being manageable for other future researchers to use without too much engineering overhead. In the future, building a better data pipeline should allow us to test Oryx at even greater scale.
>     - Oryx works very well in continuous control settings. All of the Multi-Agent MuJoCo tasks are continuous control problems, and Oryx significantly outperformed other algorithms on most of those datasets.
> * *In the sequential update scheme, how sensitive is the overall policy quality to the order in which agents are updated?*
>     - In Oryx, we randomly shuffle the order of agent updates at each learning step, similar to MAT [1] and HAPPO [2]. We agree that if you had some informed prior on which order of agent updates would be best, this may help speed up learning. However, we did not experiment with this because we wanted to keep our method as general as possible. And developing a method to automatically learn the best order of updates is out of the scope of this work, but it is an interesting direction for future work.
> * *In sequential updates, each agent optimizes assuming the policies of the other agents are fixed. Does this introduce error accumulation across the agent chain?*
>     - Indeed, the accumulation of error as agent policies are updated one after another is a central challenge in offline MARL. However, since Oryx uses an autoregressive policy scheme (like MAT [1]) where agents condition on the actions of agents that came before them, the error will be less than fully independent policies. This fact gets at the central benefit of using a model like Oryx.
> * *In your implementation of Implicit Constraint Q-learning for each agent, how is the constraint enforced during policy updates?*
>     - The ICQ constraint is enforced in two ways. First, during the critical learning step (Q-learning), one strictly only uses the actions observed in the dataset for computing the target value for the next state. This avoids extrapolation error due to out-of-distribution actions in the value function. Then, at the policy learning step, the policy distribution is updated with respect to in-distribution actions only (i.e. actions in the dataset), which implicitly avoids out-of-distribution actions. Together, these constraints mitigate the accumulation of extrapolation error. More details on the ICQ loss can be found in the original paper [3]. Central to our contribution is the sequential and autoregressive variant of the ICQ loss.
>
> **On the Ablation Study**
>
> Finally, with regard to the reviewer's comment about the lack of ablation studies, we specifically designed the TMAZE experiments to isolate which components of Oryx were important. Our main takeaway from the TMAZE ablation study was that removing any one component from Oryx (autoregressive actions, memory, or ICQ) resulted in failure on the task. Highlighting that only by combining all three ingredients together did the algorithm succeed. We conducted the ablation in our custom-designed TMAZE environment because we felt its simplicity made the results more interpretable. If, however, the reviewer feels the ablation study would be strengthened by conducting it over a more complex, but less interpretable, environment (e.g. RWARE), we would happily consider adding that in for the camera-ready version.
>
> In conclusion, thank you once again for your very constructive questions. They have helped us sharpen our messaging in the paper. In particular, we will add a discussion on scaling to the revised version, along with more detail on how exactly the autoregressive policy updates work in practice. Should the reviewer have any follow-up questions, please do not hesitate to ask. We would be happy to engage during the author-reviewer discussion period.
>
> [1] Wen, Muning, et al. “Multi-Agent Reinforcement Learning is a Sequence Modeling Problem.” NeurIPS ‘22
>
> [2] Zhong,Yifan, et al. “Heterogeneous-agent reinforcement learning.” Journal of Machine Learning Research, January ‘24
>
> [3] Yang, Yiqin, et al. “Believe What You See: Implicit Constraint Approach for Offline Multi-Agent Reinforcement Learning” NeurIPS ‘21

---

### Note · Authors · 2025-08-13

Dear AC and Reviewers,

We thank you for giving your time to review our paper. We are grateful to all the reviewers who have helped us improve the quality and clarity of our manuscript, and to the AC for considering the overall merits for publication. We are particularly grateful to the reviewer **wEyu**, who engaged with us at length. While the concerns raised were misunderstood by us at times, we truly believe that we have reached a constructive conclusion, which has improved our evaluation and strengthened our empirical claims.

To briefly summarise our main contributions again, we developed Oryx by deriving a novel autoregressive ICQ loss and integrating it into a modified sequence modelling architecture. Our method required significant architectural and algorithmic changes from what existed before, and provided substantial performance improvements compared to the existing state-of-the-art (across more than 65 diverse benchmark datasets and 14 state-of-the-art DTDE, CTDE, and CTCE offline algorithms and at scales reaching up to 50 agents), demonstrating its effectiveness.

We have presented a robust and compelling argument for the importance, significance, and novelty of our work and find that the reviews and responses to our paper reflect an overall positive assessment and understanding of the contribution it makes to the Offline MARL research community. We are therefore hopeful for a positive outcome on the final decision for publication.

We once again would like to thank the reviewers and AC for their time and for helping us produce the best version of our manuscript.

Best regards,
Authors

---

### Decision · Program_Chairs · 2025-09-17

**Decision:**

Accept (poster)

**Comment:**

This work proposes Oryx, an offline MARL algorithm, targeting the extrapolation error and miscoordination that arise when learning from fixed datasets. It combines Sable and a sequential form of implicit constraint Q-Learning (ICQ). It theoretically proves that the multi-agent joint-policy under ICQ regularisation can be optimised sequentially for an auto-regressive model. Through experiments, the authors show that their method performs better than multiple methods on multiple benchmarks.

The strengths of this work are summarized as follows.
1. It combines ICQ and the Advantage Decomposition Theorem to address the extrapolation error and miscoordination problem. Such an approach is novel.
2. Oryx demonstrates state-of-the-art results across multiple offline MARL benchmarks.
3. Introducing new large-scale datasets for community use.

The weaknesses of this work are summarized as follows.
1. While the primary contribution is a well-engineered integration of known ideas, the empirical performance and release of datasets make it a good addition to the offline MARL literature.
2. Oryx is a CTCE method with autoregressive policies, while many baselines are CTDE with independent policies. Oryx has more information (joint observations + joint actions) compared to baselines (except MADIFF).